# eIF4A supports an oncogenic translation program in pancreatic ductal adenocarcinoma

Karina Chan[1,12], Francis Robert[2,12], Christian Oertlin[3,12], Dana Kapeller-Libermann[1,12], Daina Avizonis[2], Johana Gutierrez[1], Abram Handly-Santana[1,4], Mikhail Doubrovin[5], Julia Park[4,10], Christina Schoepfer[4], Brandon Da Silva[4,11], Melissa Yao[4], Faith Gorton[4], Junwei Shi[6], Craig J. Thomas[7], Lauren E. Brown[8], John A. Porco Jr.[8], Michael Pollak[9], Ola Larsson[3]*, Jerry Pelletier[2]* & Iok In Christine Chio[1]*

Pancreatic ductal adenocarcinoma (PDA) is a lethal malignancy with limited treatment options. Although metabolic reprogramming is a hallmark of many cancers, including PDA, previous attempts to target metabolic changes therapeutically have been stymied by drug toxicity and tumour cell plasticity. Here, we show that PDA cells engage an eIF4F-dependent translation program that supports redox and central carbon metabolism. Inhibition of the eIF4F subunit, eIF4A, using the synthetic rocaglate CR-1-31-B (CR-31) reduced the viability of PDA organoids relative to their normal counterparts. In vivo, CR-31 suppresses tumour growth and extends survival of genetically-engineered murine models of PDA. Surprisingly, inhibition of eIF4A also induces glutamine reductive carboxylation. As a consequence, combined targeting of eIF4A and glutaminase activity more effectively inhibits PDA cell growth both in vitro and in vivo. Overall, our work demonstrates the importance of eIF4A in translational control of pancreatic tumour metabolism and as a therapeutic target against PDA.

[1] Institute for Cancer Genetics, Department of Genetics and Development, Columbia University Medical Center, New York, NY 10032, USA. [2] Department of Biochemistry, Oncology and Goodman Cancer Centre, McGill University, Montreal H3G 1Y6 QC, Canada. [3] Department of Oncology-Pathology, Karolinska Institute, Stockholm, Sweden. [4] Cold Spring Harbor Laboratory, Cold Spring Harbor, NY 11724, USA. [5] Department of Radiology, Columbia University Medical Center, New York, NY 10032, USA. [6] Department of Cancer Biology, University of Pennsylvania, Philadelphia, PA 19104, USA. [7] National Cancer Institute, Rockville, MD 20850, USA. [8] Department of Chemistry and Center for Molecular Discovery (BU-CMD), Boston University, Boston, MA 02215, USA. [9] Department of Medicine and Oncology, McGill University, Montreal, QC, Canada. [10] Present address: Department of Chemistry, University of Pennsylvania, Philadelphia, PA 19104, USA. [11] Present address: SUNY Downstate College of Medicine, SUNY Downstate Medical Center, Brooklyn, NY 11203, USA. [12] These authors contributed equally: Karina Chan, Francis Robert, Christian Oertlin, Dana Kapeller-Libermann. *email: Ola.Larsson@ki.se; jerry.pelletier@mcgill.ca; ic2445@cumc.columbia.edu

Pancreatic ductal adenocarcinoma (PDA), a highly aggressive malignancy with limited treatment options, is now the third leading cause of cancer death in the United States[1]. Most PDA patients share common pathogenic lesions, including a high frequency of oncogenic KRAS mutations (90–95%) and the inactivation of certain tumour suppressors, such as TP53, P16/INK4A, and SMAD4[2,3]. Indeed, genetically-engineered mouse models (GEMMs) of PDA harbouring the $Kras^{LSL-G12D}$, $Trp53^{LSL-R172H}$, and Pdx1-cre (KPC) alleles recapitulate the complex histopathological features of human PDA[4]. Nonetheless, due to its low neoplastic cellularity, it remains challenging to study the molecular mechanisms underlying PDA. Using a three-dimensional organoid culture system that allows direct comparison between primary tumour cells and their normal, proliferating counterparts[5], we previously showed that PDA is dependent on a KRAS-mediated increase in protein synthesis[6]. However, the repertoire of mRNAs that are translationally altered by oncogenic KRAS signalling remains unknown.

In most human cancers, the oncogenic signalling pathways that promote tumorigenesis also act to dysregulate protein synthesis[7]. For example, the PI3K/AKT/mTOR and MEK/Mnk signalling pathways both converge on eukaryotic initiation factor 4F (eIF4F) to promote cap-dependent mRNA translation[7,8]. It is widely acknowledged that a rate-limiting step of cap-dependent translation is the assembly of an active eIF4F complex[7], high levels of which correlate with poor prognosis and drug resistance in various cancers[7,9]. eIF4F is a heterotrimeric complex consisting of the cap-binding protein eIF4E, the large scaffolding protein eIF4G, and the DEAD-box RNA helicase eIF4A[10]. The eIF4A helicase is required to unwind cap-proximal secondary structures within the mRNA 5′-UTR during scanning as a prelude to its association with the 43S preinitiation complex, in an ATP-dependent fashion[11].

Given the diversity of downstream AKT/mTOR effectors, previous attempts to suppress translation in cancer cells therapeutically by targeting this pathway have yielded adverse phenotypes, such as compensatory increases in MAP kinase[12] or receptor tyrosine kinase signalling[13]. Moreover, inhibition of mTORC1 using rapamycin leads to increased pancreatic tumorigenesis through the induction of extracellular protein uptake[14]. Therefore, direct targeting of the eIF4F complex has recently emerged as an attractive antineoplastic strategy. It is well established that the helicase activity of eIF4A is the predominant target of the rocaglates, a family of translation inhibitors that harbour a common cyclopenta[b]benzofuran skeleton produced exclusively by the Aglaia genus of angiosperms[15–19]. Here, we show that PDA organoids exhibit heightened sensitivity to the rocaglate CR-1-31-B (CR-31) relative to normal, proliferating ductal organoids. In addition, polysome profiling identified an oncogenic translation program in PDA organoids comprised of mRNAs involved in redox and central carbon metabolism. Indeed, we observe an increase in both glycolysis and oxidative phosphorylation in PDA organoids compared to their normal counterparts. Moreover, by disrupting this translation program, treatment of PDA cells with CR-31 induces an energy crisis and cell death. In vivo, CR-31 suppresses pancreatic cancer cell growth and improves the survival of pancreatic tumour-bearing mice. Thus, our study uncovers a previously unrecognized role for translation in PDA metabolic reprogramming, and provides a framework for targeting tumour-specific translation programs in the clinic.

## Results

**eIF4A is a therapeutic target in PDA.** We previously showed that protein synthesis is elevated in pancreatic cancer cells in a redox-dependent manner[6]. Indeed, using a non-isotopic method to measure total in vivo protein synthesis (SUnSET)[20], we observed markedly increased labelling of pancreatic ductal carcinoma cells relative to normal ductal cells (Supplementary Fig. 1a). Using murine pancreatic organoid cultures grown from ductal isolates of wildtype, normal (N), and $Kras^{G12D}$;$p53^{R172H}$-bearing tumour (KP) tissues[5,6], we also detected elevated incorporation of O-propargyl-puromycin (OP-puro)[21] into nascent polypeptides after a 30 min pulse in KP compared to N organoids (Supplementary Fig. 1b). Thus, protein synthesis is hyperactive in both organoid and in vivo models of PDA. These observations support the notion that cancer cells require an aberrantly activated translational state for survival and suggest that targeting translation might provide a favourable therapeutic index for pancreatic cancer[22]. To explore this possibility in PDA, we evaluated selective inhibition of the DEAD-box RNA helicase, eIF4A, using the rocaglate, CR-31 (Supplementary Fig. 1c)[16].

Consistent with previous studies of rocaglates[16,23,24], CR-31 substantially decreases the off-rate of eIF4A from polypurine-RNAs in the presence of ATP or AMP-PNP (Supplementary Fig. 1d), thus confirming its inhibitory effect on eIF4A. When N and KP organoids were treated with CR-31 for 1 h, we found that N organoids were minimally affected (Fig. 1a), while a striking dose-dependent decrease in nascent protein synthesis was observed in KP organoids as measured by either [35S]-methionine (Supplementary Fig. 1e) or OP-puro (Fig. 1a) incorporation. Consistent with this observation, KP organoids exhibit higher sensitivity to CR-31 compared to N organoids in viability assays (Fig. 1b, Supplementary Fig. 1f), with a 10 fold decrease in half maximal inhibitory concentration ($IC_{50}$) (Supplementary Figs. 1f, g). CR-31 is also cytotoxic towards a panel of patient-derived PDA cell lines, with similar $IC_{50}$ values (Supplementary Fig. 1g, h). These results suggest that CR-31 is a potent inhibitor of PDA protein synthesis and survival in vitro.

Encouraged by our in vitro observations, we tested the effects of CR-31 using a murine orthotopic transplant model of PDA. When tumour diameters reached 7–9 mm by ultrasound imaging, mice were randomly assigned into treatment groups with either CR-31 or vehicle, following a dosage regimen previously reported to exert no toxicity towards cells of hematopoietic origin[16]. At this dose, CR-31 effectively inhibited protein synthesis (Fig. 1c) and growth (Fig. 1d) of pancreatic tumours. The effect of CR-31 was cytotoxic instead of cytostatic, as evidenced by a 5 fold increase in immuno-staining of cleaved caspase 3 (Fig. 1e) but not phospho-histone H3 (Supplementary Figs. 1i, j). Importantly, this dosage regimen did not induce measurable changes in body weight (Supplementary Fig. 2a) or in circulating levels of liver (Supplementary Fig. 2b) or pancreatic (Supplementary Fig. 2c) enzymes. Gender-balanced cohorts of non-tumour-bearing C57Bl/6J mice given the same treatment for 12 days also displayed no changes in body weight (Supplementary Fig. 2d), circulating levels of liver enzymes (Supplementary Fig. 2e) or tissue histology (Supplementary Fig. 2f). In contrast to pancreatic tumours (Fig. 1e), CR-31 treatment did not induce caspase 3 cleavage in the healthy pancreas even after 12 days of treatment (Supplementary Fig. 2g). Finally, in the autochthonous KPC model[4], CR-31 improved survival compared to vehicle-treated controls (Fig. 1f). Collectively, these results suggest that CR-31-mediated eIF4A inhibition can suppress PDA growth without inducing measurable systemic toxicity in vivo. eIF4A may therefore represent an attractive therapeutic target for PDA.

**CR-31 modulates mRNA-selective translation in PDA.** We next sought to decipher the mechanisms underlying the enhanced sensitivity of PDA to CR-31. Given that the PI3K/AKT/mTOR and MEK/MNK/eIF4E signalling pathways can regulate both

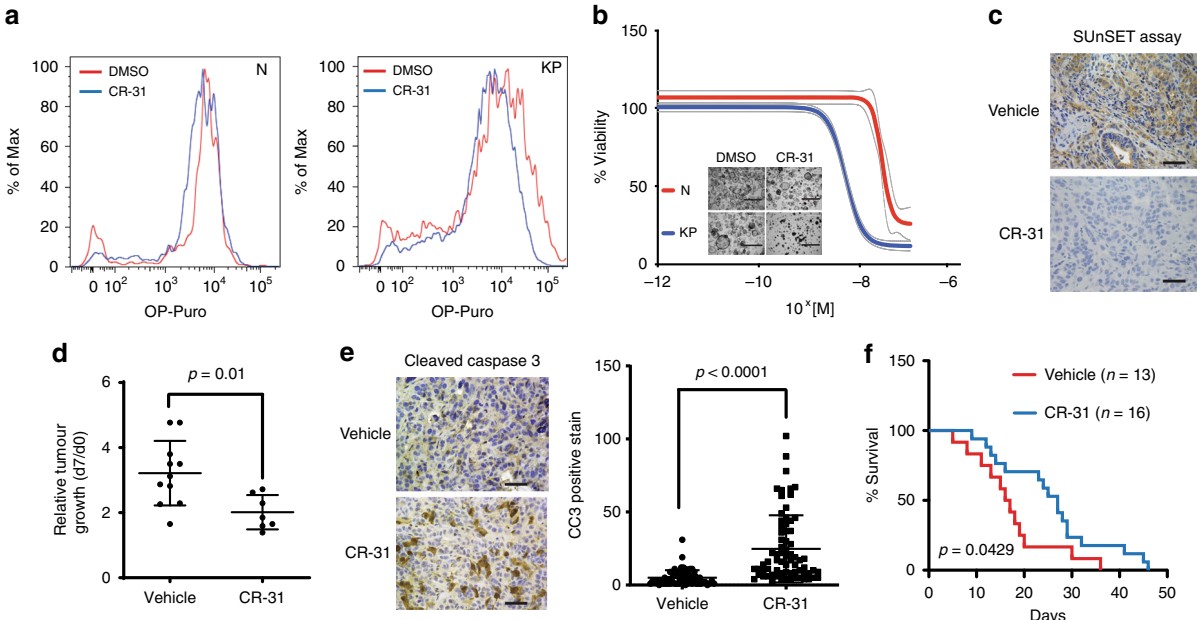

**Fig. 1** The eIF4A inhibitor CR-31 potently targets pancreatic ductal adenocarcinoma. **a** Normal (N), and $Kras^{G12D};p53^{R172H}$ (KP) organoids were treated with 10 nM CR-31 for 60 min and compared to vehicle (DMSO). During the last 30 min, 10 µM O-propargyl-puromycin (OP-Puro) was supplemented into the media. Dissociated single cells were fixed, stained by CuAAC with Alexa488-azide and the level of OP-Puro conjugation within polypeptide chains was quantified by flow cytometry. **b** Cell viability of N and KP organoids upon a 72 h treatment with increasing concentrations of CR-31. Dotted lines indicate 95% confidence intervals ($n = 5$). Inset shows representative images of organoids treated with 10 nM CR-31 for 72 h. Scale bars, 1000 µm. **c** SUnSET assay of representative PDA tumours from mice following 7 daily treatments with vehicle (5.2% PEG-400/5.2% Tween80) or 0.2 mg kg$^{-1}$ CR-31. Mice were euthanized for tissue collection 2 h after the final treatment with vehicle or CR-31 and 30 min after 40 nmol g$^{-1}$ puromycin injection. Scale bars, 50 µm. **d** Relative tumour volume of PDA-bearing mice after 7 daily treatments with 0.2 mg kg$^{-1}$ CR-31 or vehicle. Relative growth was estimated by comparison of tumour volume 7 days post injection to that at pre-enrolment (day 0) by ultrasound imaging. Data are mean +/- S.D. Student's t-test. ($n = 11$, vehicle; $n = 7$, CR-31). **e** Immunohistochemistry of cleaved caspase 3 (left) in representative tumours from mice treated daily for 7 days with vehicle or 0.2 mg kg$^{-1}$ CR-31 together with quantification of cleaved caspase 3 (right) from $n \geq 7$ fields of view. Data are mean +/-S.D. Student's t-test. Tissues were harvested 2 h post final treatment. Scale bars, 50 µm. **f** Kaplan–Meier survival analysis for $Kras^{G12D};p53^{R172H};PdxCre$ (KPC) mice treated daily with vehicle ($n = 13$) or 0.2 mg kg$^{-1}$ CR-31 ($n = 16$). Mantel-Cox (log rank) test. Source data are provided as a Source Data file

global and mRNA-selective translation via modulation of eIF4F-complex formation[25–27], we first examined whether signalling through these pathways can be modulated by CR-31 treatment. We did not observe quantitative decreases in the phosphorylation status of proteins in the PI3K (AKT, mTOR, S6, 4EBP) and MEK (eIF4E) cascades in either murine (Supplementary Fig. 3a, b, d) or human (Supplementary Fig. 3c) PDA cells after 1 h or 5 h of CR-31 treatment. Interestingly, N organoids exhibited an increase in phospho-S6 at both time points, but without a corresponding change in phospho-4EBP1 levels (Supplementary Fig. 3a, d). We also noted a transient induction of eIF4E phosphorylation in CR-31-treated KP organoids which subsided at 5 h post treatment (Supplementary Fig. 3b, d), though these changes were not observed in patient-derived PDA cell lines (Supplementary Fig. 3c). These results indicate that inhibition of translation upon CR-31 treatment is not due to an indirect suppression of the PI3K/AKT/mTOR or MEK/MNK/eIF4E signalling pathways. Notably, acute CR-31 treatment did not induce phosphorylation of eIF2α (Supplementary Fig. 3e), thereby precluding a primary role for the integrated stress response in CR-31-mediated suppression of translation[28].

The eIF4F complex sits at the junction of several potent oncogenic pathways including, most notably, c-MYC. As translation of c-MYC mRNA is highly eIF4F-dependent[7], we asked whether the cytotoxic effect of CR-31 on pancreatic tumours in vivo is due to a reduction of c-MYC expression. While CR-31 treatment suppresses tumour growth in vivo (Fig. 1d), these tumours, collected 2 h after final treatment, do not exhibit a

decrease in c-MYC protein levels (Supplementary Fig. 3f). Thus, we reasoned that reduced c-MYC expression is not a driver of the tumour suppressive activity of CR-31 in PDA cells. Therefore, we asked whether selective alterations in the translation efficiency of other transcripts might underlie the observed differences in response to CR-31. To this end, we performed polysome profiling to compare the effects of CR-31 on the translatomes (i.e., the pool of efficiently translated mRNAs) of N and KP organoids (Supplementary Fig. 4a). During polysome profiling, efficiently translated mRNAs (associated with ≥3 ribosomes) were isolated and quantified. Anota2seq[29] was used to compare the changes in polysome-associated mRNAs to the total transcript levels of each mRNA. Given that altered levels of total mRNA will also impact the pool of polysome-associated mRNA, this allows the identification of bona fide changes in translation efficiency. Through this analysis, we identified strong differences between the translatomes of vehicle-treated N and KP organoids (Fig. 2a). Consistent with the oncogenic activity of the eIF4F complex in cancer cells, transcripts previously shown to exhibit eIF4E-dependent translation[30] (eIF4E-signature) are translationally activated in KP relative to N organoids (Supplementary Fig. 4b). Interestingly, while CR-31 treatment has minimal impact on the N translatome (Fig. 2b, Supplementary Data 1), it induces marked changes in the KP translatome (Fig. 2c, Supplementary Data 2). Consistently, the eIF4E-signature is translationally suppressed by CR-31 only in KP (Supplementary Fig. 4c), but not in N organoids (Supplementary Fig. 4d). Strikingly, upon CR-31 treatment, the subset of mRNAs that are translationally activated

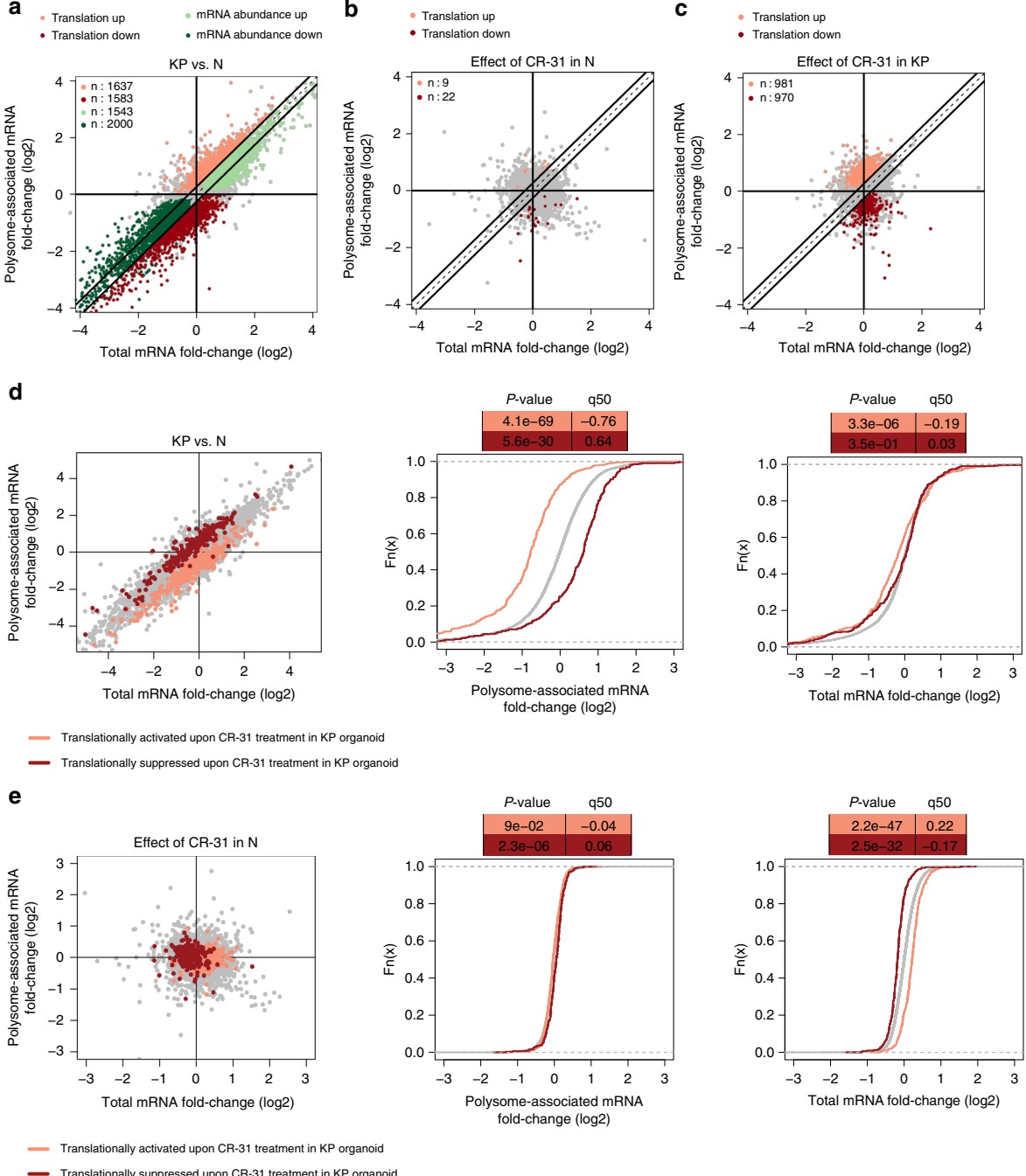

**Fig. 2** Tumour-selective modulation of translational efficiencies by CR-31. **a** Scatterplot of polysome-associated mRNA to total mRNA log2 fold-changes comparing vehicle-treated $Kras^{G12D};p53^{R172H}$ (KP) to normal (N) organoids. The numbers of mRNAs with a change in translation efficiency (light red and dark red) or mRNA abundance (light green and dark green) are indicated ($n = 3$; positive log2 fold-change indicates higher translation/abundance in KP organoids). **b** Scatterplot of polysome-associated mRNA to total mRNA log2 fold-changes comparing N organoids treated with vehicle or 10 nM CR-31 for 1 h. The numbers of mRNAs with a change in translation efficiency (light red and dark red) are indicated (negative log2 fold-change indicates suppressed translation upon CR-31 treatment). **c** Scatterplot of polysome-associated mRNA to total mRNA log2 fold-changes comparing KP organoids treated with vehicle or 10 nM CR-31 for 1 h. The numbers of mRNAs with a change in translation efficiency (light red and dark red) are indicated (negative log2 fold-changes indicates suppressed translation upon CR-31 treatment). **d** Scatterplot of log2 fold-changes for polysome-associated mRNA and total mRNA (left), an empirical cumulative distribution (ecdf) of log2 fold-changes for polysome-associated mRNA (middle) and total mRNA (right) comparing KP and N organoids. Transcripts whose translation was activated (light red) or suppressed (dark red) upon CR-31 treatment in KP organoids (i.e. from (**c**)) are indicated for each plot. Wilcoxon rank-sum test p-values comparing regulation of subsets to background are indicated together with the log2 fold-changes difference between these subsets to the background at the 50th percentile (q50) of the ecdf. **e** Same representation as in panel **d** but examining differences in gene expression in N organoids upon CR-31 treatment. Source data are provided as a Source Data file

in KP relative to N organoids revert towards their translational state in N organoids, such that the translatome of CR-31-treated KP organoids resembles that of vehicle-treated N organoids (Fig. 2d, Supplementary Data 3). In contrast, translation of this mRNA subset is unaffected upon CR-31 treatment in N organoids (Fig. 2e). In line with our earlier observation, c-MYC is not amongst the mRNAs that are sensitive to CR-31 in N or KP organoids.

Factors required for efficient translation of an mRNA are determined in large part by diverse features of its 5′ untranslated region (UTR), including its length, secondary structure, and sequence motifs. Therefore, we next sought to identify RNA features associated with the effects of CR-31 on mRNA translation. Consistent with a recent report examining the 5′-UTRs of eIF4A-dependent mRNAs[31], we did not observe an enrichment for $(GGC)_4$ motifs, which are proposed to fold into G-quadruplexes, within the 5′-UTRs of mRNAs whose translation is suppressed by CR-31 (Supplementary Fig. 5a). We also did not observe an enrichment for polypurines[24] in the 5′-UTRs of transcripts that are translationally suppressed in KP organoids after CR-31 treatment (Supplementary Fig. 5b). Instead, we found that transcripts which are negatively regulated by CR-31 treatment have slightly shorter 5′-UTRs (Supplementary Fig. 5c), exhibit an elevated GC content (Supplementary Fig. 5d), and are more structured (after adjustment of fold-energies for 5′-UTR lengths) than non-regulated transcripts (Supplementary Fig. 5e). Thus, the RNA helicase eIF4A supports an oncogenic translation program that is activated in pancreatic cancer cells and characterized by mRNAs with structured 5′-UTRs. Overall, our results indicate that while inhibition of eIF4A by CR-31 exhibits minimal impact on the translatome of N organoids, it effectively reprograms the translatome of KP organoids towards that of N organoids.

**PDA cells enhance translation of redox metabolism enzymes.**
To determine whether the subset of proteins whose synthesis is targeted by CR-31 in KP organoids represents particular biochemical processes, we performed KEGG pathway enrichment analysis using the gage R package[32]. Consistent with a previous study showing that eIF4E is critical for selective translation of mRNAs that control intracellular levels of reactive oxygen species (ROS) during cellular transformation[33], we observed translational activation of mRNAs involved in glutathione metabolism in KP relative to N organoids (Fig. 3a, Supplementary Fig. 6). Importantly, the translation of several of these mRNAs is selectively downregulated in KP organoids upon CR-31 treatment (Fig. 3a, Supplementary Fig. 6). These results suggest that maintenance of redox homeostasis is an oncogenic activity of the eIF4F complex that can be targeted using CR-31.

The ratio of the ROS scavenger glutathione (GSH) to its oxidized form (GSSG) is a marker of oxidative stress. We found that several key enzymes involved in glutathione synthesis, as well as the production of NADPH (the cofactor that supports regeneration of GSH from GSSG), are translationally induced in KP organoids, and suppressed upon CR-31 treatment (Fig. 3a). Indeed, CR-31 treatment led to a significant decrease in NADPH, and a corresponding increase in the ratio of $NADP^+$ to NADPH in KP, but not N (Fig. 3b) organoids. Similar responses were observed in patient-derived PDA cell lines (Fig. 3b). Treatment of these PDA lines with CR-31 also led to a dose-dependent decrease in total GSH levels (Fig. 3c) and in the ratio of GSH to GSSG (Fig. 3d). Likewise, CR-31 treatment moderately increased the levels of ROS in KP organoids (Fig. 3e). The increase in ROS was also indicated by the accumulation of Nuclear factor erythroid-derived 2-like 2 (NFE2L2/NRF2), a central regulator of redox

control[34] (Fig. 3f), and transactivation of its downstream target genes (Fig. 3g). While genetic ablation of Nrf2 did sensitize KP organoids to CR-31 treatment (Fig. 3h), direct inhibition of glutathione synthesis using buthionine sulfoximine (BSO) did not phenocopy the effect of Nrf2 ablation (Fig. 3i). The ROS scavenger N-acetyl-cysteine (NAC) also failed to alleviate the cytotoxicity of CR-31 in these cells (Fig. 3j). Thus, although eIF4A inhibition using CR-31 perturbs redox homeostasis, the cytotoxic effect of CR-31 is unlikely to be mediated by redox dysregulation.

**PDA translation program supports central carbon metabolism.**
Relative to the translatome of N organoids, we found that the most highly translated mRNAs in KP organoids encode proteins involved in oxidative phosphorylation (Supplementary Fig. 6). In fact, almost every protein component of the electron transport chain (ETC) is encoded by transcripts that are translationally activated in KP organoids (Fig. 4a), despite no apparent changes in mRNA levels (Fig. 4b). For example, expression of the Sdhb protein, a subunit in complex II, is upregulated in KP relative to N organoids (Supplementary Fig. 7a). In accordance with the translational activation of mRNAs encoding ETC components in KP organoids, we observed that at steady state, oxidative phosphorylation is elevated in KP relative to N organoids, as determined by the rate of mitochondrial oxygen consumption (OCR) (Supplementary Fig. 7b) and the percentage contribution of uniformly-labelled $^{13}C_6$-glucose into citrate in the TCA cycle (Fig. 4c). Strikingly, KEGG pathway analysis shows that the translation of ETC transcripts is suppressed by CR-31 in KP organoids (Fig. 4a, Supplementary Fig. 6). We further confirmed by quantitative RT-PCR that mRNAs encoding proteins in complex I and II (Ndufa3 and Sdhb) show decreased polysome association in CR-31-treated KP organoids (Supplementary Fig. 7c). Although mitochondrial numbers are largely unaffected by CR-31 treatment, as determined by MitoTracker Green staining (Supplementary Fig. 7d), OCR was markedly decreased in a dose-dependent manner upon 5 h of CR-31 treatment in both KP organoids (Fig. 4d) and patient-derived PDA cell lines (Supplementary Fig. 7e), but to a much lower extent in N organoids (Fig. 4e). The resultant energy stress is also evident from the induction of phospho-AMPK in KP organoids (Supplementary Fig. 7f). In accordance with the polysome profiling data, expression of the Saccharomyces cerevisiae NADH dehydrogenase (ETC complex I) NDI1[35] only partially rescued CR-31-induced suppression of oxidative phosphorylation in PDA cells (Supplementary Fig. 7g), indicating that CR-31 impacts multiple complexes of the ETC (Fig. 4a). Importantly, shRNA-mediated depletion of eIF4A1 also suppressed oxidative phosphorylation in both murine (Supplementary Fig. 7h) and human (Supplementary Fig. 7i) PDA cells, confirming that the availability of eIF4A1 plays a role in the regulation of oxidative phosphorylation.

Defective oxidative phosphorylation is expected to induce a metabolic shift towards aerobic glycolysis. As a consequence, cells treated with inhibitors of oxidative phosphorylation generally exhibit a heightened sensitivity to glucose deprivation[36]. However, we observed that PDA cells were not more sensitive to CR-31 under low glucose conditions (Supplementary Fig. 8a). This prompted us to investigate the impact of eIF4A inhibition on glycolysis in greater detail. Comparing N and KP organoids, we observed that KP organoids exhibit elevated levels of glycolysis, as determined by the extracellular acidification rate (ECAR) (Supplementary Fig. 8b) and the contribution of uniformly-labelled $^{13}C_6$-glucose to lactate after 2 h of culture in $^{13}C_6$-glucose (Supplementary Fig. 8c). The relative partitioning of glucose label between citrate in the TCA cycle and lactate, as determined by the ratio of lactate M + 3 to citrate M + 2, is also increased in KP

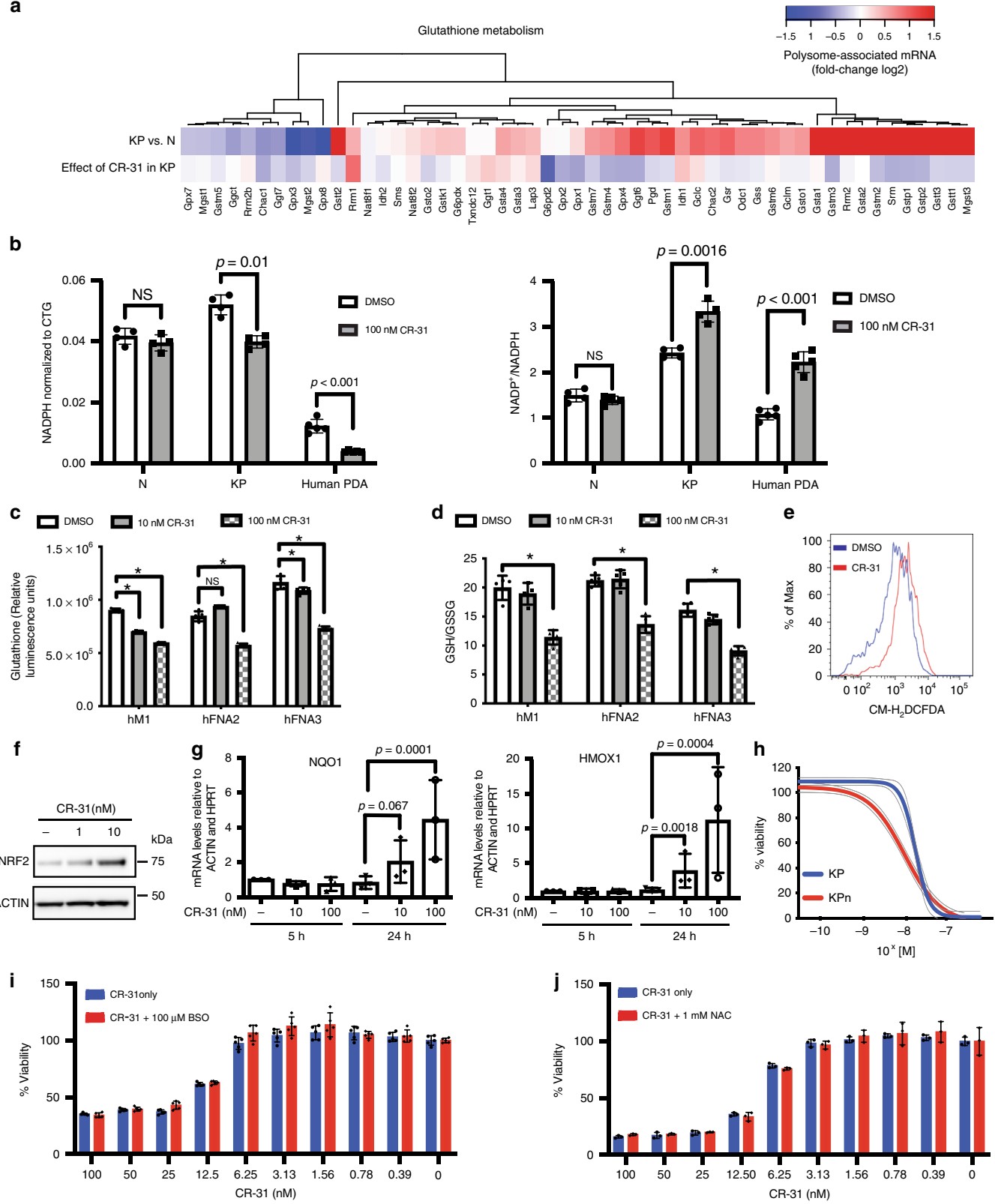

organoids (Supplementary Fig. 8d). When KP organoids (Fig. 4f, Supplementary Fig. 8e, f) or patient-derived PDA cells (Fig. 4f) were treated with CR-31, global levels of intracellular and secreted lactate were reduced. Interestingly, this effect was much more potent in KP organoids than in N organoids (Fig. 4f). Consistent with the reduction of glycolysis in CR-31-treated PDA cells, we observed a decreased contribution of $^{13}C_6$-glucose in all detectable upstream intermediates of the glycolytic pathway (Supplementary Fig. 8g). In an acute glycolytic stress test, CR-31 treatment for 5 h suppressed the ECAR of KP organoids (Fig. 4g)

**Fig. 3** CR-31 translationally targets glutathione metabolism in PDA. **a** Heatmap of glutathione metabolism pathway proteins (KEGG database) showing fold-changes of polysome-associated mRNA. **b** Relative levels of NADPH normalized to viability based on CellTitre-Glo (CTG) assay (left) and ratio of NADP$^+$ to NADPH (right) in organoids and patient-derived PDA lines upon treatment with vehicle (DMSO) or 100 nM CR-31 for 16 h. Data are mean +/−S.D. Student's $t$-test, $n = 4$ (organoids), $n = 5$ (patient-derived lines). NS = not significant. **c** Relative levels of glutathione in patient-derived PDA lines upon treatment with vehicle or CR-31 for 24 h. Data are mean +/−S.D., Student's $t$-test. $n = 5$, NS = not significant. *$p < 0.05$ compared to DMSO treatment. **d** Ratio of GSH to GSSG in patient-derived PDA lines upon treatment with vehicle or the indicated concentrations of CR-31 for 24 h. Data are mean +/−S.D., Student's $t$-test, $n = 5$. *$p < 0.05$ compared to DMSO treatment. **e** ROS levels in KP organoids upon treatment with vehicle or 10 nM CR-31 for 16 h. Data are representative from three biological replicates. **f** NRF2 protein levels in KP organoids treated with CR-31 for 16 h. Actin, loading control. Representative image from 3 biological replicates. **g** mRNA expression of NRF2 target genes in 3 patient-derived PDA lines upon treatment with CR-31 for 5 or 24 h. Data are mean +/−S.D. Each data point plotted is average from every biological replicate. **h** Cell viability of KP and *Nrf2*-deficient KP (KPn) organoids upon a 72 h treatment with increasing concentrations of CR-31. Dotted lines indicate 95% confidence intervals, $n = 5$. **i** Cell viability of patient-derived PDA lines upon treatment with increasing concentrations of CR-31 in the presence or absence of 100 μM buthionine sulfoximine (BSO) for 72 h. Data are mean +/−S.D., $n = 5$. **j** Cell viability of primary murine PDA lines upon treatment with increasing concentrations of CR-31 in the presence or absence of 1 mM of the ROS scavenger N-acetylcysteine (NAC) for 72 h. Data are mean +/−S.D., $n = 3$. Source data are provided as a Source Data file

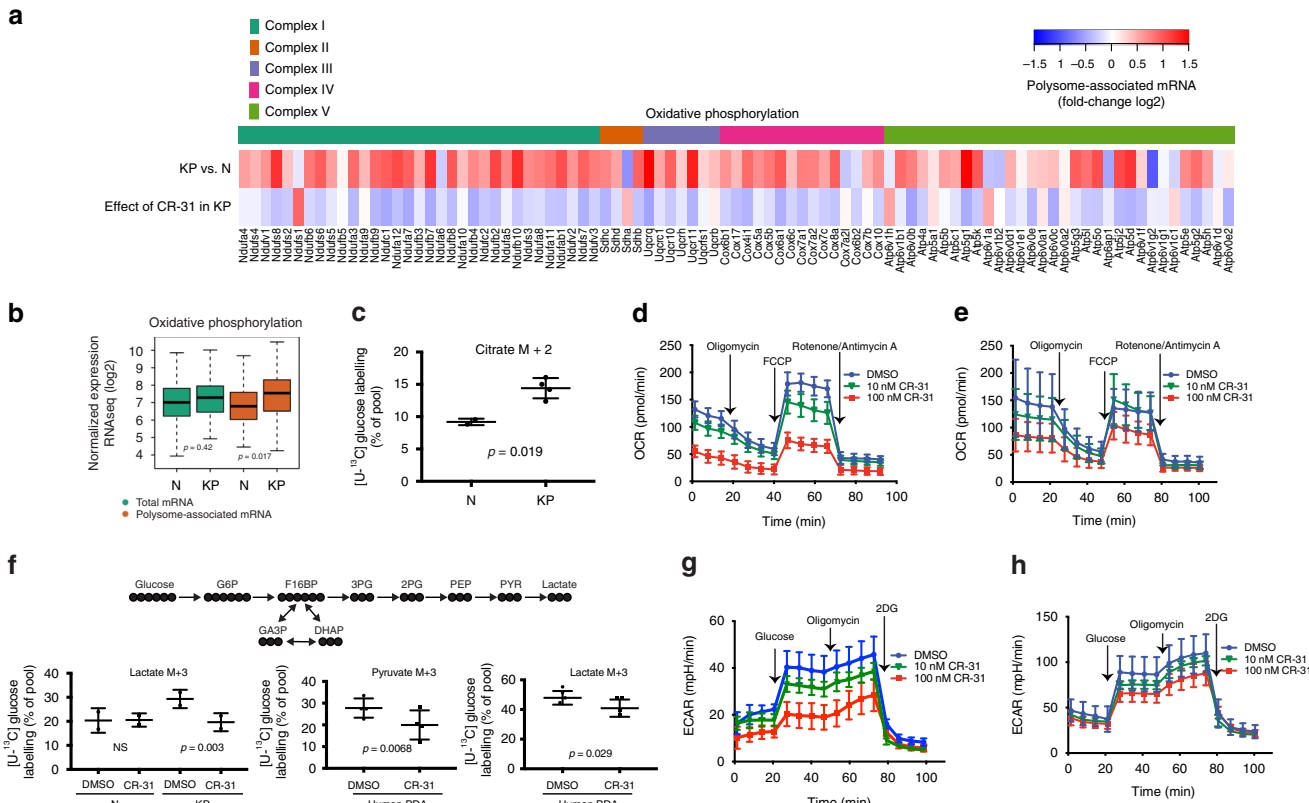

**Fig. 4** CR-31 translationally targets oxidative phosphorylation and glycolysis in PDA. **a** Heatmap of oxidative phosphorylation pathway proteins showing fold-changes of polysome-associated mRNA. **b** Boxplots of total and polysome-associated mRNA levels for transcripts encoding oxidative phosphorylation components. Horizontal line indicates the median; lower and upper box indicate the first and third quartiles. Whiskers extend from the box to the most extreme value but not further than +/−1.5 * inter-quartile range (IQR) from the box. Data exceeding the whiskers are plotted as points. **b** $^{13}C_6$-glucose tracing of N ($n = 2$) and KP organoids ($n = 4$). M +2 labelled citrate is derived directly from $^{13}C_6$-glucose. Data are mean +/− S.D., Student's $t$-test. **c** Mitochondrial respiration reflected by oxygen consumption rate (OCR) was measured in KP organoids treated with vehicle or CR-31 for 5 h and following the addition of Oligomycin (1 μM), the uncoupler FCCP (0.5 μM), and the electron transport inhibitor Rotenone/Antimycin A (0.5 μM). Data are mean +/− S.D, $n = 3$. Data are representative from three biological replicates. **d** OCR was measured in N organoids treated with vehicle CR-31 for 5 h and following the addition of Oligomycin, FCCP, and Rotenone/Antimycin A. Data are mean +/− S.D. $n = 5$. Data are representative from three biological replicates. **e** Top, flow of heavy carbons through the glycolytic pathway. Bottom, $^{13}C_6$-glucose tracing of two sets of N and KP organoids (left) and 4 patient-derived PDA cell lines (middle and right) upon treatment with vehicle or 100 nM CR-31 for 5 h. Data are mean +/− S.D., Student's paired $t$-test. **f** Glycolysis reflected by extracellular acidification rate (ECAR) was measured in KP organoids treated with vehicle or CR-31 for 5 h and following addition of glucose (10 mM), Oligomycin (1 μM), and the glucose analog, 2-deoxyglucose, 2DG (50 mM). Data are mean +/− S.D., $n = 3$. Data are representative of three biological replicates. **g** ECAR was measured in N organoids treated with vehicle or CR-31 for 5 h and following addition of glucose, Oligomycin, and 2DG. Data are mean +/− S.D., $n = 5$. Data are representative of three biological replicates. Source data are provided as a Source Data file

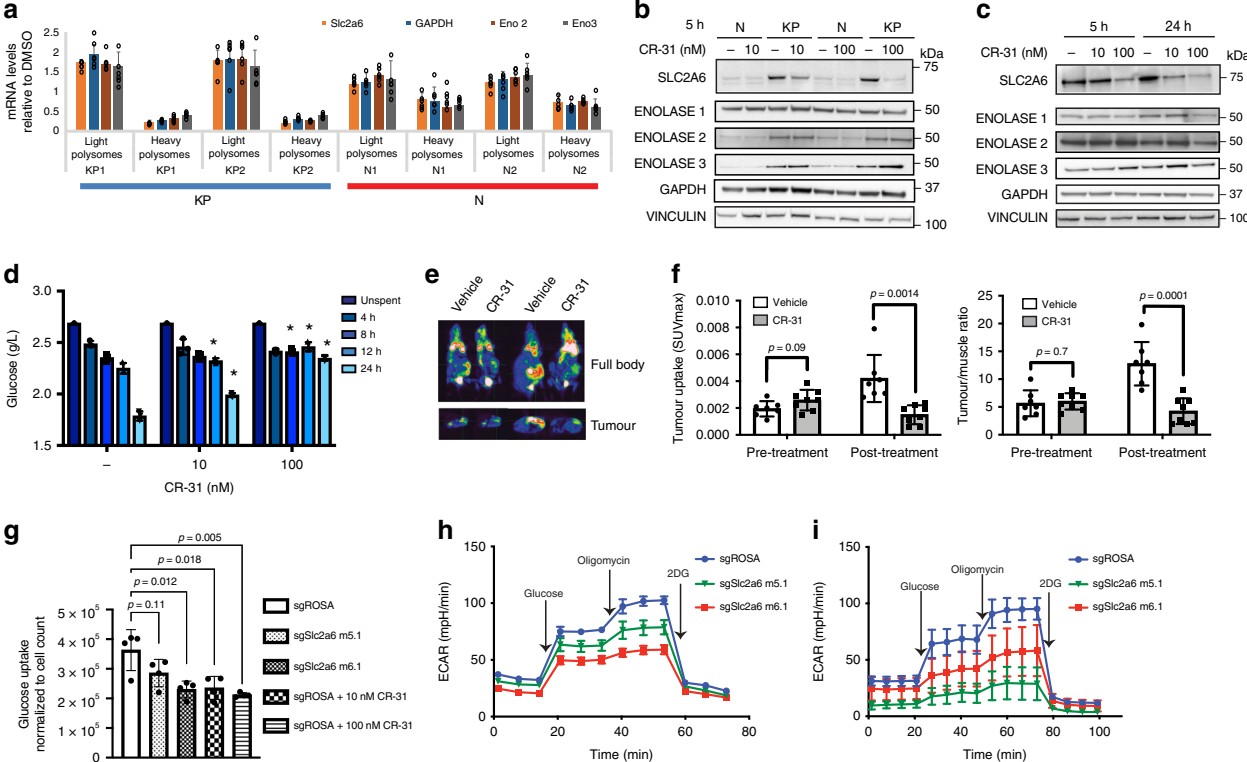

**Fig. 5** CR-31 translationally targets glucose uptake in PDA. **a** Levels of transcripts encoding glycolytic proteins in light (<3 ribosomes) or heavy (≥3 ribosomes) polysome fractions from N or KP organoids treated with 10 nM CR-31 for 1 h relative to vehicle. Data are mean +/− S.D. $n = 6$. **b** Immunoblot analysis of SLC2A6, ENOLASE 1, ENOLASE2, ENOLASE 3, and GAPDH in N and KP organoids upon vehicle or CR-31 treatment for 5 h. Vinculin, loading control. **c** Immunoblot analysis of SLC2A6, ENOLASE1, ENOLASE2, ENOLASE3, and GAPDH in patient-derived PDA lines upon vehicle or CR-31 treatment. VINCULIN, loading control. **d** Glucose concentration in media spent from KP organoids upon treatment with vehicle or CR-31. Data are mean +/− S.D., Student's t-test, $n = 3$. *$p < 0.05$, compared to DMSO treatment at the same timepoint. **e** Representative $^{18}$F-FDG-PET images of pancreatic tumours from mice after 24 h of treatment with vehicle or CR-31. Top, coronal view; bottom, transverse view. **f** Tumour uptake plots comparing initial metabolic activity before and after treatment with vehicle or CR-31. Left, maximum standardized uptake value (SUV)$_{max}$ of tumour. Right, ratio of (SUV)$_{max}$ in tumour compared to muscles. Data are mean +/− S.D., Student's t-test. $n = 7$, Vehicle; $n = 8$, CR-31. **g** Glucose uptake comparing Cas9-expressing murine PDA cell lines transduced with sgRNAs targeting the *ROSA* locus (sg*ROSA*) or against different regions of the *Slc2a6* gene (sg*Slc2a6*). Data are mean +/− S.D., normalized to cell numbers. Student's t-test, $n = 4$. **h** ECAR was measured in equal numbers of Cas9-expressing primary murine PDA cell lines transduced with sgRNAs targeting the *ROSA* locus (sg*ROSA*) or different regions of the *Slc2a6* gene (sg*Slc2a6*). ECAR was measured following the addition of glucose, Oligomycin, and 2DG. Data are mean +/− S.D., $n = 5$. **i** ECAR was measured in equal numbers of Cas9-expressing KP organoids transduced with sgRNAs targeting the *ROSA* locus (sg*ROSA*) or different regions of the *Slc2a6* gene (sg*Slc2a6*). ECAR was measured following the addition of glucose, Oligomycin, and 2DG. Data are mean +/− S.D., $n = 6$. Source data are provided as a Source Data file

and patient-derived PDA cell lines (Supplementary Fig. 8h) in a dose-dependent manner. CR-31 also impacted N organoids in an acute glycolytic stress test, but to a lower extent (Fig. 4h).

Given the above observations, we first examined the expression levels of various glycolytic enzymes. Since CR-31 treatment up to 24 h did not consistently decrease the expression of these enzymes at either the mRNA (Supplementary Fig. 9a) or the protein level (Supplementary Fig. 9b), it is unlikely that CR-31 suppresses glycolysis by perturbing the expression of these proteins. As oncogenic KRAS maintains PDA by regulating anabolic glucose metabolism[37,38], we also asked whether CR-31 impairs glycolysis by directly suppressing KRAS levels. However, neither murine (Supplementary Fig. 10a, b) nor human (Supplementary Fig. 10c, d) PDA cells exhibit a decrease in KRAS expression level following CR-31 treatment. In fact, there seems to be a transient induction of KRAS expression after 1 h of CR-31 treatment, which subsides at later timepoints (Supplementary Fig. 10a–d). Collectively, our data indicate that the metabolic alterations occurring upon CR-31 treatment cannot be attributed to suppression of either KRAS or the glycolytic proteins examined.

Upon manual inspection of our polysome profiling data, we found several enzymes involved in glucose transport and metabolism to be targets of CR-31. For example, in KP organoids, but not N organoids, a 1 h treatment with 10 nM CR-31 markedly reduced the levels of polysome-associated mRNAs encoding the glucose transporter *Slc2a6*, and the glycolytic enzymes *GAPDH*, *Enolase 2*, and *Enolase 3* (Fig. 5a). Of these, however, only the SLC2A6 glucose transporter displayed a clear decrease in steady-state protein levels after CR-31 treatment in KP organoids (Fig. 5b, Supplementary Fig. 11a) and patient-derived PDA cell lines (Fig. 5c). This is likely due to the relatively shorter half-life of this transporter compared to the long-lived GADPH and ENOLASE enzymes (Supplementary Fig. 11b). In accordance with this observation, we found that CR-31 treatment decreased glucose uptake in a dose-dependent fashion (Fig. 5d). These results were validated in vivo by FDG-PET analysis of PDA orthotopic transplants, which revealed a marked reduction in tumour glucose uptake relative to muscle after CR-31 treatment as compared to vehicle controls (Fig. 5e, f).

Elevated glucose uptake in cancer cells is traditionally attributed to the glucose transporters SLC2A1 and SLC2A3[39].

However, these transporters were not identified as CR-31-sensitive transcripts by polysome profiling. Consistently, minimal changes were observed in the expression of these transporters in whole cell lysates (Supplementary Fig. 9b) and at the membrane level (Supplementary Fig. 11c, d). We therefore asked if impaired glucose metabolism in CR-31 treated cells might be attributed to the less characterized glucose transporter, SLC2A6. To evaluate the contribution of SLC2A6 to glucose uptake in PDA cells, we perturbed *Slc2a6* expression using two independent sgRNAs (Supplementary Fig. 11e, f). Compared to cells expressing a control sgRNA targeting the ROSA locus, sg*Slc2a6* expression decreased glucose uptake (Fig. 5g) and glycolysis in murine PDA cell lines (Fig. 5h) and in KP organoids (Fig. 5i). Collectively, our data suggest that CR-31 suppresses glycolysis in PDA cells in part by suppressing the translation of the mRNA encoding glucose transporter SLC2A6. As a consequence, CR-31 inhibition of SLC2A6 translation blunts the compensatory glycolytic program that is normally engaged by cancer cells subjected to ETC inhibition[36].

**CR-31 induces a dependency on reverse glutaminolysis.** As described above, CR-31 treatment of PDA cells inhibits translation of multiple factors implicated in oxidative phosphorylation and glycolysis. However, polysome profiling also identified a distinct subset of mRNAs that are translationally upregulated upon CR-31 treatment in KP organoids (Fig. 2c). While many of these mRNAs are involved in diverse developmental processes, we noted that some of these transcripts encode enzymes involved in glutamine transport, such as *Slc1a5*[40], and glutamine metabolism, such as *Gls1*[41]. Indeed, quantitative RT-PCR confirmed an increase in the association of these mRNAs with polysomes upon 1 h of CR-31 treatment in KP but not in N organoids (Fig. 6a). Consistently, expression of the GLS1 protein is increased in patient-derived PDA cell lines upon CR-31 treatment (Fig. 6b).

We observed that CR-31 increases the intracellular levels of glutamine in a panel of patient-derived PDA cell lines (Fig. 6c), likely due to translational upregulation of its transporter SLC1A5 (Fig. 6a). To better understand the metabolic fate of glutamine upon eIF4A inhibition, we cultured PDA cells in $^{13}C_5$ glutamine-containing media for 2 h and quantified downstream metabolites using GC/MS. As described earlier, CR-31 treatment targets the synthesis of ETC components to compromise oxidative mitochondrial function (Fig. 4a). Consistent with this, we noted a decrease in oxidative labelling of metabolites downstream of glutamine, including malate, fumarate, and aspartate (Fig. 6d). However, we found a significant increase in reductive labelling of citrate and cis-aconitate from $^{13}C_5$ glutamine in both CR-31-treated human (Fig. 6e) and murine PDA cells (Fig. 6f), reflecting a shift towards reductive carboxylation, during which glutamine-derived α-ketoglutarate is carboxylated to produce isocitrate/citrate, which is then cleaved to generate oxaloacetate and acetyl-CoA[42]. Notably, this shift to reductive carboxylation was not observed in N organoids given the same treatment (Fig. 6f). The pathway of reductive carboxylation engages NADPH-dependent isocitrate dehydrogenase, and subsequent metabolism of glutamine-derived citrate provides both the acetyl-CoA for lipid synthesis and the four-carbon intermediates needed to produce the remaining TCA metabolites and related macromolecular precursors[43]. This reductive, glutamine-dependent pathway has been shown to be the dominant mode of metabolism in rapidly growing malignant cells containing mutations in ETC complexes I or III, and in cells with normal mitochondria subjected to acute pharmacological ETC inhibition[44].

The increased glutamine reductive carboxylation of CR-31-treated cells prompted us to ask if combined inhibition of eIF4A and glutamine metabolism might represent a more effective therapeutic strategy than eIF4A inhibition alone. Indeed, we found that the glutaminase inhibitors BPTES[45] and CB839[46] (Fig. 6g, h) readily sensitize PDA cells to CR-31. Gemcitabine, one of the main chemotherapeutics used to treat pancreatic cancer, did not elicit the same effect (Supplementary Fig. 12a), demonstrating the selective sensitivity of CR-31-treated cells to the inhibition of glutaminase. Given that glutamine is a precursor for glutathione synthesis[47], and that CR-31 treatment perturbs glutathione metabolism (Fig. 3), we initially asked whether glutaminase inhibition sensitized PDA cells to CR-31 through further induction of oxidative stress. However, CB839 treatment did not increase ROS levels (Supplementary Fig. 12b) or NADP$^+$ to NADPH ratios (Supplementary Fig. 12c) above those induced by CR-31 alone. Accordingly, the ROS scavenger NAC failed to mitigate the heightened sensitivity of PDA cells to CR-31 by CB839 (Supplementary Fig. 12d). Given that glutamine-derived citrate, in the process of reductive carboxylation, provides acetyl-CoA for lipid synthesis[43], we tested the effect of Orlistat, a fatty acid synthesis inhibitor[48], and surprisingly found that it was able to weakly sensitize patient-derived PDA cell lines to CR-31 (Supplementary Fig. 12e). These results suggest that the sensitization of PDA cells to CR-31 through co-suppression of glutaminase activity occurs through redox-independent mechanisms, possibly by reducing carbon flow towards fatty acid synthesis. Using PDA organoid/cancer-associated fibroblast co-cultures, we observed a therapeutic synergy between CR-31 and CB839 (Fig. 6i). In vivo, combined inhibition of eIF4A and glutaminase exhibited an additive effect and suppressed pancreatic tumour growth in an orthotopic transplant model (Supplementary Fig. 12f). Together, these data reveal that eIF4A is critical for PDA maintenance and delineate an eIF4A-dependent translation program that promotes redox and central carbon metabolism in pancreatic cancer cells.

**Discussion**

Since cancer cells often acquire an aberrantly activated translational state, inhibition of translation initiation should target neoplastic cells with limited toxicity for normal cells[22]. Indeed, our previous work demonstrated that protein synthesis is elevated in pancreatic cancer cells in a redox-dependent manner[6]. However, the full repertoire of mRNAs which are translationally altered by oncogenic signalling and the underlying molecular mechanisms that govern their translational control remain poorly understood. In this study, we demonstrate that PDA cells have a heightened dependency on the function of RNA helicase eIF4A. Moreover, using transcriptome-wide polysome profiling, we uncovered an oncogenic translation program composed of a specific subset of transcripts that are sensitive to eIF4A availability, and can be suppressed upon treatment with the rocaglate CR-31.

While ours and previous studies[24,49,50] support the notion that modulation of eIF4A activity leads to selective effects on translation, the underlying mechanisms are still incompletely understood. It has been suggested that G-quadruplexes in 5′-UTRs mediate selective suppression of translation downstream of eIF4A inhibition by the rocaglate silvestrol[50], while eIF4A clamping to polypurine sequences underlie the effects of the rocaglate Roca-glamide A[24]. Our findings suggest that in PDA cells slightly shorter 5′-UTRs with more stable secondary structures contribute to the selective suppression of translation following CR-31 treatment. Importantly, we observed that shRNA depletion of eIF4A phenocopies the metabolic changes induced by CR-31. In conjunction with our finding that polypurine sequences are not enriched in mRNAs translationally suppressed by CR-31, we

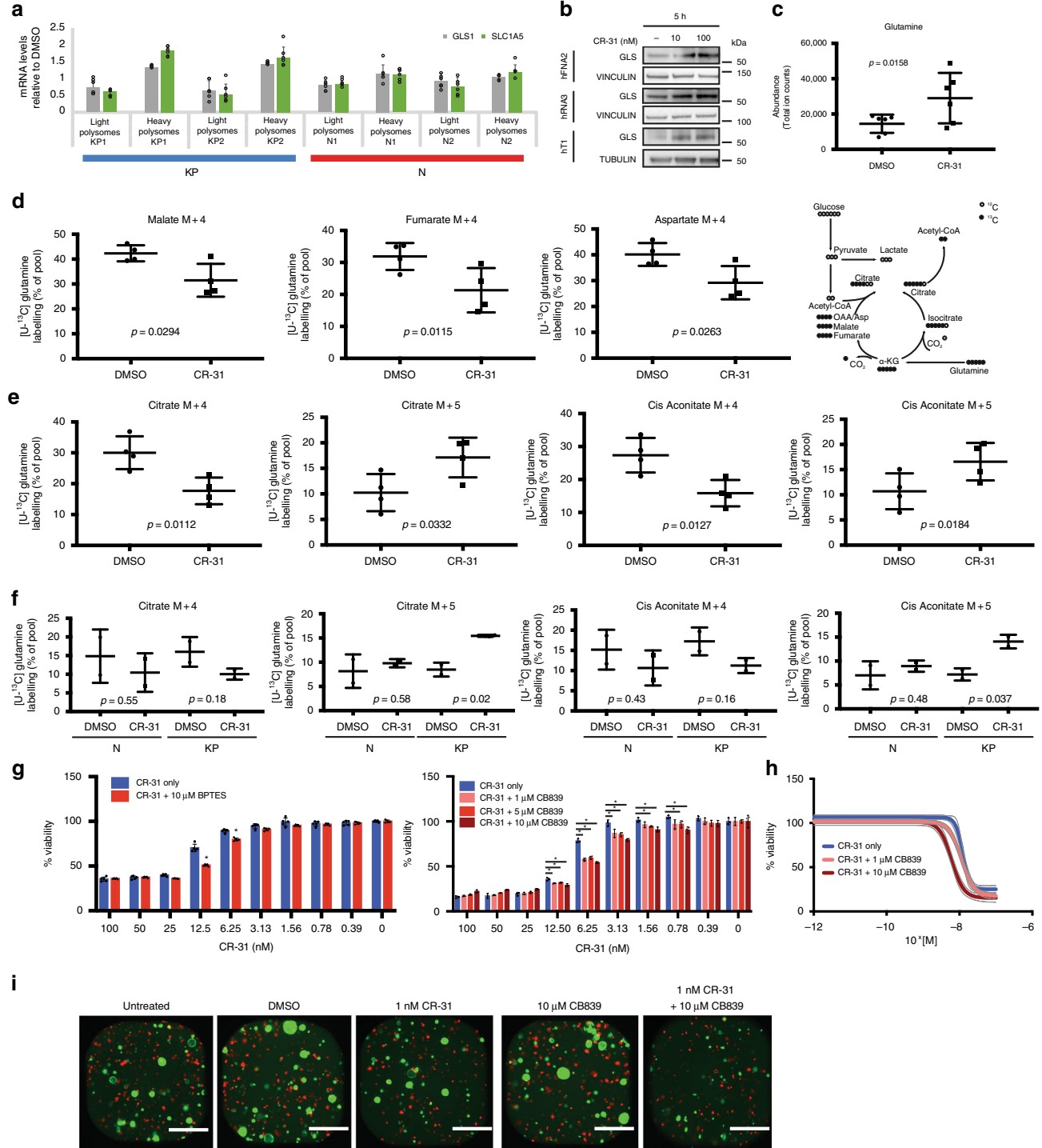

conclude that eIF4A clamping on polypurine sequences cannot account for the full effect of CR-31 on translatomes observed herein. Instead, ours and previous studies collectively indicate that rocaglates act via multiple mechanisms in a drug and/or context-dependent manner. This is further supported by our observation that the impact of CR-31 is much more potent on the translatome of KP organoids than their normal counterparts. The heightened sensitivity of the PDA translatome to CR-31 may also reflect transcriptional differences between PDA cells and normal ductal cells, which in turn alter the ability of specific mRNAs to compete for access to limiting amounts of eIF4F.

In this study, we found that the functional gene classes whose translation are strongly dependent on eIF4A in PDA include

those involved in central carbon metabolism. As first described by Otto Warburg, cancer cells often exhibit a high rate of glycolysis. Recent work further emphasizes that, in addition to a strong dependence on glycolysis, many tumours, including PDA, also rely on mitochondrial respiration for bioenergetic[51–53] and bio-synthetic processes[54–56]. These observations have prompted the use of biguanides such as Metformin, which can inhibit complex I of the ETC, as a therapeutic strategy for pancreatic cancer[57]. However, Metformin yielded disappointing results in a placebo-controlled clinical trial in pancreatic cancer[58]. Interestingly, Metformin has been shown to increase glycolysis even in cancer cells that are already highly glycolytic[59], thus attenuating its anti-neoplastic activity. Our study reveals an oncogenic translation

**Fig. 6** CR-31 treatment increases reverse glutamine metabolism in pancreatic cancer cells. **a** Levels of transcripts encoding proteins involved in glutamine metabolism in light (<3 ribosomes) or heavy (≥3 ribosomes) polysome fractions from N or KP organoids treated with 10 nM CR-31 for 1 h relative to vehicle (DMSO). Data are mean +/− S.D., $n = 6$. Data from two biological replicates are shown. **b** Immunoblot analysis of GLS in patient-derived PDA cell lines upon treatment with vehicle or CR-31 for 5 h. VINCULIN, TUBULIN, loading controls. **c** Intracellular levels of glutamine at steady state from 3 patient-derived PDA cell lines (2 technical replicates each) treated with vehicle or 100 nM CR-31 for 5 h. Data are mean +/− S.D., Student's paired $t$-test. **d** $^{13}C_5$-glutamine tracing of 4 patient-derived PDA cell lines upon treatment with vehicle or 100 nM CR-31 for 5 h. Data are mean +/− S.D., Student's paired $t$-test. Right, schematic illustrating the flow of heavy carbons through glutamine metabolic pathways. **e** $^{13}C_5$-glutamine tracing of 4 patient-derived PDA cell lines upon treatment with vehicle or 100 nM CR-31 for 5 h. M + 4 indicates metabolites of oxidative glutaminolysis and M + 5 indicates metabolites of reductive glutaminolysis. Data are mean +/− S.D., Student's paired $t$-test. **f** $^{13}C_5$-glutamine tracing of N ($n = 2$) and KP ($n = 2$) organoids upon treatment with vehicle or 100 nM CR-31 for 5 h. Data are mean +/− S.D., Student's $t$-test. **g** Cell viability of patient-derived PDA lines upon a 72 h treatment with increasing concentrations of CR-31 and the indicated concentrations of the glutaminase inhibitor BPTES ($n = 5$) or CB839 ($n = 3$). Data are mean +/− S.D., Student's $t$-test, $*p < 0.05$ compared to CR-31 only treatment. **h** Cell viability of KP organoids upon a 72 h treatment with increasing concentrations of CR-31 and the indicated concentrations of CB839. Dotted lines indicate 95% confidence intervals, $n = 5$. **i** KP organoids (GFP-labelled) in co-culture with pancreatic cancer-associated fibroblasts (mCherry-labelled) upon a 72 h treatment with CR-31 in the presence or absence of CB839. Representative images from two biological replicates. Scale bars, 1000 μm. Source data are provided as a Source Data file

program supporting both oxidative phosphorylation and glycolysis in pancreatic cancer cells. As such, inhibition of this translation program using CR-31 inhibits both oxidative phosphorylation and glycolysis in PDA cells to effectively induce energy crisis and cell death. Thus, this compound overcomes the limitations of metabolic inhibitors such as biguanides by blunting the compensatory increase in glycolysis.

We observed that the suppression of glycolysis by CR-31 occurs at the level of glucose uptake. Elevated glucose uptake in cancer cells is traditionally attributed to the glucose transporters SLC2A1 and SLC2A3[39]. In fact, an earlier study using the rocaglate RHT also reported suppression of glucose transport through increased TXNIP mRNA levels in cancer cells[60], which acts as a negative regulator of SLC2A1[61]. However, we observed no measurable change in SLC2A1 expression upon CR-31 treatment. Instead, we show that translation of the mRNA encoding a less characterized glucose transporter, SLC2A6, is suppressed by CR-31. This suggests that cancer cells may utilize non-canonical means of glucose uptake to sustain anabolic needs. While genetic knockout of this transporter in mice has minimal effect on whole body metabolic physiology[62], the expression of this transporter has been reported to be elevated in various cancers, including PDA[63,64]. In this study, we show that genetic targeting of SLC2A6 expression reduces both glucose uptake and glycolysis, supporting a functional role for SLC2A6 in PDA glucose metabolism. Our study suggests that SLC2A6 contributes in part to the glycolytic defects that arise from pharmacological inhibition of eIF4A in pancreatic cancer cells. Thus, further characterization of the contribution of SLC2A6 to glucose uptake relative to SLC2A1/3 may illuminate our understanding of cancer cell metabolism.

It is interesting to note that suppression of eIF4A by CR-31 is also associated with increased translation of a subset of mRNAs. While similar observations have been reported in prior studies involving genetic or pharmacological perturbations of the eIF4F complex[26,30,49,50], the functional relevance of these changes remains poorly understood. Here, we show that some transcripts that are translationally activated upon CR-31 treatment may constitute a metabolic compensatory mechanism to increase glutaminolysis through the reductive pathway in PDA cells. This intriguing observation supports the possibility that metabolic plasticity can occur at the level of translation. While examples of transcriptional adaptation and genetic compensation are well documented in the literature, little is known about such compensatory mechanisms at the level of translation, particularly in the context of tumour stress responses. These findings add an important dimension to the mechanisms by which KRAS drives tumorigenesis. In summary, we have delineated the importance of eIF4A in the translational control of pancreatic tumour

metabolism and illustrated how this dependency can be therapeutically exploited.

## Methods

**Animals.** $Trp53^{+/LSL-R172H}$, $Kras^{+/LSL-G12D}$, $Pdx1$-$Cre$, and $Nrf2^{-/-}$ strains in C57Bl/6J background were interbred to obtain $Pdx1$-$Cre$; $Kras^{+/LSL-G12D}$; $Trp53^{+/LSL-R172H}$ (KPC)[4] or $Pdx1$-$Cre$; $Kras^{+/LSL-G12D}$; $Trp53^{+/LSL-R172H}$; $Nrf2^{-/-}$ (KPCn) mice for therapeutic studies and for organoid isolation. C57Bl/6J mice were purchased from the Jackson Laboratory for the isolation of N organoids. All animal experiments were conducted in accordance with procedures approved by the IACUC at Columbia University (AC-AAAT7469).

**Human specimens.** All human experiments were approved by the ethical committee of Columbia University (IRB-AAAR6773). Written informed consent from the donors for research use of tissue in this study was obtained prior to acquisition of the specimen. Samples were confirmed to be tumour or normal based on pathological assessment.

**Chemical reagents.** The rocaglate (-)-CR-1-31-B (bioactive enantiomer) was synthesized as previously described[16].

**Orthotopic engraftment in the pancreas.** Mice were anesthetized using isoflurane, and Ketoprofen/Carprofen (5 mg kg$^{-1}$), which was subcutaneously administered. 2 × 2 mm tumour pieces derived from KPC were transplanted per pancreas. The abdominal wall was sutured with absorbable vicryl sutures (Ethicon Cat# J392H), and the skin was closed with wound clips (CellPoint Scientific Inc. Cat# 203-1000).

**Therapeutic intervention studies in mice.** Upon detection of a mass during weekly palpation, KPC or allograft-bearing mice were subjected to high-contrast ultrasound imaging using the Vevo 2100 System with a MS250, 13–24 MHz scanhead (Visual Sonics, Inc, Amsterdam, NL). Mice with tumour diameters of 7–9 mm were randomized and enroled one day after scanning. CR-31 was formulated in 5.2% PEG-400 (Sigma-Aldrich Cat# 1546445)/5.2% Tween80 (Sigma-Aldrich Cat#P4780) in saline. Mice were administered vehicle, 0.2 mg kg$^{-1}$ CR-31 (intraperitoneal injection, daily) and/or 200 mg kg$^{-1}$ CB839 formulated in 25% hydroxypropyl-β-cyclodextrin (Roquette Cat#3461111) (oral gavage, twice daily). Tumour volume was monitored on days 4 and 7 after initial scan. The same dosing regimen was applied to C57Bl/6 J mice bearing orthotopic mouse tumour allografts.

**Assessment of CR-31 in vivo toxicity.** Gender-balanced C57Bl/6J mice were randomized and enroled into study. CR-31 was formulated in 5.2% PEG-400/5.2% Tween80 in saline. Mice were administered vehicle or 0.2 mg kg$^{-1}$ CR-31 (intraperitoneal injection) once per day over 12 days. Body weight of each mouse was recorded prior to every dose. Mice were euthanized 2 h after the last treatment. Full necropsy was performed on the mice, organs were embedded in paraffin and sectioned and stained for H&E to look for any abnormalities. Blood was collected through cardiac puncture into a multivette to isolate plasma (Braintree Scientific Cat# MV-600 151673) for measurement of circulating aspartate aminotransferase (AST), alanine aminotransferase (ALT), amylase, and lipase levels. For accurate measurements, plasma was diluted in saline, 1:5 dilution for AST and ALT assays, and 1:2 dilution for amylase and lipase assays. AST, ALT, amylase, and lipase assays were run on the Element DC machine (Heska).

**Organoid isolation and culture.** Detailed procedures to isolate and propagate murine and human, normal and neoplastic pancreatic organoids have been described previously[5,65]. In brief, organoids were maintained in complete organoid media: Advanced DMEM/F12 (Gibco Cat#12634010) supplemented with 1% Penicillin/Streptomycin (PS), 1x GlutaMAX (Gibco Cat# 35050061), 1x Hepes (Gibco Cat# 15630080), 1x B27 (Invitrogen Cat# 17504044), 1.25 mM N-Acetylcysteine (NAC) (Sigma-Aldrich Cat# A9165), 10 nM gastrin (Sigma-Aldrich Cat# G9145), 50 ng/ml EGF (PeproTech Cat# 315-09), 10% RSPO1-conditioned media, 20% Noggin-conditioned media, 100 ng/ml FGF10 (PeproTech Cat# 100-26), and 10 mM Nicotinamide (Sigma-Aldrich Cat# N0636). To passage, organoids were washed out from the GFR-Matrigel (Corning Cat# 356231) using cold Advanced DMEM/F12 supplemented 1% PS, 1x GlutaMAX and 1x Hepes, and mechanically dissociated into small fragments using fire-polished glass pipettes, and then seeded into fresh GFR-Matrigel. Passaging was performed at a 1:4 split ratio roughly twice per week. All experiments described were done in the absence of EGF and NAC[6]. Human cultures were characterized by sequencing DNA to confirm that they harbour loci representative of human PDA, and the mouse organoids were genotyped to ensure they carry mutant alleles of *Kras*, *Trp53* and/or *Nfe2l2*. Once the cells are cultured in our laboratories, they are regularly screened for mycoplasma contamination and authenticated by short tandem repeat (STR) analysis to identify misidentified, cross-contaminated, or genetically drifted cells.

**Cell culture conditions for monolayer cultures.** Monolayer KPC primary pancreatic cancer cells, patient-derived pancreatic cancer cells, and murine cancer-associated fibroblasts (CAFs) were cultured in DMEM (Gibco Cat#11995073) supplemented with 10% fetal bovine serum (FBS) (Corning Cat# 35-010-cv) and 1% PS (Gibco Cat# 15140163). All cells were cultured at 37 °C with 5% $CO_2$.

**Co-culture conditions.** For co-cultures, GFP-labelled murine organoids were split at a 1:2 ratio and mixed with $1 \times 10^4$ mCherry-labelled CAFs, seeded in 50 μl GFR-Matrigel, and cultured in DMEM supplemented with 10% FBS and 1% P/S.

**Therapeutic experiments with organoids.** CR-31, CB839 (Selleckchem Cat# S7655), and Orlistat (Selleckchem Cat# S1629) were dissolved in DMSO (Sigma-Aldrich Cat# D2650). The final concentration of DMSO was no higher than 0.2%. Buthionine sulfoximine (Sigma-Aldrich Cat# B2640) was dissolved in PBS. Nine different doses and a vehicle were used for the cytotoxicity assays for CR-31 as indicated in each figure legend. When CR-31 was used in combination with CB839, the dose of CB839 was fixed to 1, 5 or 10 μM. When CR-31 was used in combination with Orlistat, the dose of Orlistat was fixed to 50 μM. When CR-31 was used in combination with BSO, the dose of BSO was fixed to 100 μM. For $IC_{50}$ analysis, organoids were dissociated to single cells by triturating organoids in media through a fire-polished glass pipette, and then by enzymatic dissociation with 2 mg ml$^{-1}$ dispase (Gibco Cat# 17105041) dissolved in TrypLE (Life Technologies Cat# 12604013) at 37 °C, until the organoids appeared as single cells under the microscope (~15 min). Cells were counted, and diluted to 30 cells μl$^{-1}$ in a mixture of complete media minus EGF and NAC, and GFR-Matrigel (10% final concentration). 100 μl of this mixture (3000 cells per well) was plated in a 96 well plate (Corning Cat# 3917) previously coated with a bed of GFR-Matrigel (20 μl per well, 1:1 PBS mixture). Once organoids reformed (between 24–36 h post-plating, confirmed by microscopy), drugs were added in 100 μl of media. As described above, nine different doses plus a vehicle control were used for each drug, and at least five replicate wells were treated with each dose. 72 h after the addition of the drug, cell viability was measured by area covered using CellInSight CX5, or using a luminescence ATP-based assay (CellTiter-Glo, Promega Cat# G7573) with a plate reader (SpectraMax i3x, Molecular Devices).

**Analysis of global protein synthesis.** Isotope labelling: Organoids were incubated for 30 min in media containing 10 μCi of [$^{35}$S] labelled Methionine (Perkin Elmer Cat# NEG709A005MC). For liquid scintillation, equal amounts of radiolabelled total protein were TCA-precipitated and washed two times in acetone and air dried at room temperature. The amount of [$^{35}$S]-Met incorporated into protein was measured using a Beckman LS6500 Scintillation counter. Total protein content was determined by BCA assay (Bio-Rad Cat# 5000116).

O-propargyl-puromycin labelling: Click-iT assays were performed using $1 \times 10^6$ cells per assay according to the manufacturer's protocol. In brief, O-propargyl-puromycin (OP-Puro; 20 μM) (Life Technologies Cat# C10456,) was added to the cells and incubated for 30 min. Cells were washed in ice-cold PBS and then fixed and permeabilized. The amount of alexa-Fluor-488 conjugated OP-Puro was quantified using flow cytometry using a BD LSR Fortessa Cytometer (BD Biosciences). Data analysis was performed using FlowJo v9.7.6 (FlowJo, Ashland, RRID:SCR_008520).

**SUnSET (Surface Sensing of Translation) assay.** The in vivo SUnSET assay was adapted from a published protocol[20]. Puromycin (40 nmol g$^{-1}$ body weight) (Sigma-Aldrich Cat# P9620) was injected intraperitoneally in mice 30 min prior to their sacrifices. The pancreas was then removed, fixed in 10% formalin for one day, embedded in paraffin and sliced. Immunohistochemistry was performed using

1:1000 dilution of the anti-puromycin 12D10 antibody (Millipore Cat# MABE343, RRID:AB_2566826).

**Polysome fractionation and qRT-PCR.** Organoids were treated with vehicle or 10 nM CR-31 for 1 h. In the final 5 min, 100 μg ml$^{-1}$ of Cycloheximide was supplemented into the media. The organoids were then harvested on ice in PBS containing 100 μg ml$^{-1}$ Cycloheximide. Cells were pelleted and lysed in 10 mM Tris-HCl (pH 8), 140 mM NaCl, 1.5 mM MgCl$_2$, 0.1% Triton X-100, 50 mM DTT, 150 μg ml$^{-1}$ Cycloheximide, and 640 U ml$^{-1}$ RNasin (Sigma Cat# 3335399001) for 15 min. Lysates were cleared, and then loaded onto a 10–50% sucrose gradient made using a Biocomp Gradient Master 108 and centrifuged for 2 h and 15 min at $151,263 \times g$ in a SW41 rotor using a Sorvall Discovery 90SE. The gradients were fractionated on a Teledyne ISCO Foxy R1 apparatus while monitoring the $OD_{254}$.

**RNA-sequencing and bioinformatics analysis.** Fractions corresponding to mRNA populations bound by ≥3 ribosomes were pooled. RNA was extracted using TRIzol (Invitrogen Cat# 15596018) according to the manufacturer's instructions. Prior to loading of the sample on the sucrose gradient, a fraction of the lysate was also taken for RNA extraction, corresponding to the total RNA fraction. The RNA was then submitted to RNA-Seq on an Illumina HiSeq 4000 PE100 at the McGill University and Genome Quebec Innovation Center.

RNA-Seq reads were subjected to and passed quality control (Phred scores > 30) and were subsequently aligned to the hg38 reference genome using HISAT2 (version 2.04) with default settings. Reads mapped to multiple locations in the genome were discarded. Gene expression was quantified using the RPKMforgenes.py method (settings -fulltranscript, -onlycoding)[66] from which raw gene counts were obtained (version last modified 11.04.2013). Genes that could not be resolved (based on sequence similarity) and/or had 0 counts in at least one sample were removed from the analysis. Genes were annotated using the RefSeq database. A total of 11402 unique genes were represented in the RNAseq data. Non-normalized RNAseq data was used as input in anota2seq (ver. 1.2.0; with parameters datatype = "RNAseq", normalize = TRUE, transformation = "TMM-log2"). Changes in translational efficiencies altering protein levels were assessed using the anota2seqAnalyze function. Contrasts were set to assess the effect of CR-31 in N organoids, the effect of CR-31 in KP organoids and the difference between non-treated N and KP organoids. Genes were considered significantly regulated when passing filtering criteria (parameters for anota2seqSelSigGenes function) [maxPAdj (FDR) <0.25], [selDeltaPT >log2(1.2)], [minSlopeTranslation >−1], [maxSlopeTranslation <2], [selDeltaTP >log2(1.2)], [minSlopeBuffering >−2] and [maxSlopeBuffering <1], [selDeltaP >log2(1)], [selDeltaT >log2(1)]. Changes in mRNA abundance were then assessed using the anota2seqRegModes function. 5′-UTR sequences were obtained from the RefSeq database (mm10)[67]. If multiple sequences for a gene were present only the longest was taken into consideration. 5′-UTR GC content and length were calculated based on the obtained sequences. Linear regression of 5′-UTR energy ~5′-UTR length was performed to obtain residuals representing free energy levels of 5′-UTRs adjusted for 5′-UTR lengths. KEGG pathway[68] enrichment analysis was performed using -log10(FDR) * sign (log2FC) of polysome-associated mRNA changes and the gage R-package[32] (version 2.30.0, saaTest = ts.KStest). The datasets generated during and/or analysed during the current study are available in the GEO repository under the accession number **GSE125380**.

**RNA-Seq validation.** For the RNA-Seq validation, organoids were treated with CR-31 and polysomes were fractionated as described above. The fractions were pooled into two groups corresponding to mRNA populations that were bound to either 3 ribosomes and up or 2 ribosomes or less. The RNA was extracted using TRIzol and cDNA was made using M-MuLV Reverse Transcriptase (New England Biolabs Cat# M0253S) and oligo(dT)20 primers. qPCRs were performed with SsoFast Evagreen Supermix (Bio-Rad Cat# 1725200) using the CFX96 PCR system (Bio-Rad Cat# 1855195). The following primers were used: *Gls1*: For: 5′ GGGCA TGATGTGTTGGTCTC 3′, Rev: 5′ ATGCGGCAAACAGAAGGTTT 3′; *Slc1a5*: For: 5′ TTTCAGCCTCCATTCTCGGT 3′ Rev: 5′ TGGAATCCACCGCTACTA CC 3′; *GAPDH*: For: 5′ AACGGATTTGGCCGTATTGG 3′ Rev: 5′ CATTCTCG GCCTTGACTGTG 3′; *Enolase 2*: For: 5′ AGAAGGCCTGCAACTGTTTG 3′ Rev: 5′ TGTACACAGTCCGACGACAA 3′; *Enolase 3*: For: 5′ GGGTCCCTCTCTAC CGACAC 3′ Rev: 5′ ATGCGCATGGCTTCCTTGAA 3′; *Sdhb*: For: 5′ GAATGC AGACGTACGAGGTG 3′; Rev: 5′ CAAGAGCCACAGATGCCTTC 3′; *Ndufa3*: For: 5′ TGCCTTCCTCAAGAATGCCT 3′; Rev: 5′ GCACTGGGTAGTTGTAG GGT 3′. *Slc2a6*: 5′ TGTACCGGCCTGTTCTCATT 3′ CCAACTATCGCTGCA TCCTG. The presence of each mRNA upon CR-31 treatment relative to DMSO in the two groups was calculated using the ΔCT method.

**Flow cytometry.** Mitochondrial numbers and cellular ROS: For cellular reactive oxygen species (ROS) and mitochondrial staining, cells were labelled in phenol-red-free Advanced DMEM/F12 media containing 10 μM 2′,7′-dichlorofluorescein diacetate (Invitrogen Molecular probes Cat# C6827) or 100 nM Mito-tracker Green FM (Cell Signalling Technology Cat# 9074), respectively, for 30 min, and analysed by flow cytometry.

SLC2A1 surface expression: Dissociated single cells were fixed with 4% paraformaldehyde at room temperature for 15 min. Cells were incubated with SLC2A1 antibody (Abcam, Cat# 40084, RRID: AB_2190927), 1:100 dilution in 0.5% BSA/PBS, followed by anti-mouse Alexa Fluor488 (Thermo Fisher Scientific Cat# R10477) in 0.5% BSA/PBS, and analyzed by flow cytometry.

**Lentiviral and retroviral production and infection**. pLenti-Cas9, LRC2.1T-sgRNA and human pPrime-PGK-Puro shRNA lentiviruses were produced in 293 T cells co-expressing the packaging vectors (pPAX2, Addgene# 12260 and VSVG, Addgene# 12259), concentrated with LentiX concentrator (Clontech Cat# 631232), and resuspended with DMEM supplemented with 10%FBS and 1% PS at 5× concentration. Mouse MLS shRNA retroviruses were produced in Phoenix-ECO cells (ATCC Cat# CRL-3214, RRID:CVCL_H717), concentrated with RetroX Concentrator (Clontech Cat# 631456), and resuspended in DMEM supplemented with 10% FBS and 1% PS, or Advanced DMEM/F12 (Gibco Cat#12634010) supplemented with 1% Penicillin/Streptomycin (PS), 1× GlutaMAX (Gibco Cat# 35050061), and 1× Hepes (Gibco Cat# 15630080) at 5× concentration. To knock-down the expression of eIF4A1, short hairpin RNAs against eIF4A1 were used. For mouse cell lines, shRenilla was used as a control, while she*IF4A1*.371 (5′-ATTGA TATGGCAAATGTAGCTG-3′) and she*IF4A1*.372 (5′-AATTGATATGGCAAAT GTAGCT-3′). For human cell lines, sh*Renilla* was used as a control, and she*IF4A1*.969 was used to knockdown eIF4A1 (5′-TCGAGCATGCATCTTCTCG GTG-3′). Two hundred thousand cells were plated and infected with 5× concentrated virus with 5 μg ml⁻¹ polybrene (Millipore Sigma Cat# TR1003G) and spinoculated at 600 RCF for 45 min at room temperature. For organoids, spinoculated cells were kept at 37 °C with 5% CO₂ for 6 h prior to replating in Matrigel and organoid media. One day after infection, pPrime-PGK-Puro shRNA infected cells were treated with 2 μg ml⁻¹ puromycin (Sigma-Aldrich Cat# P9620) for selection.

**CRISPR/Cas9-mediated gene deletion**. Organoid lines were transduced with lentivirus expressing Cas9 (EFS-Cas9-P2A-Puro, Addgene: 108100). In this study, all sgRNAs targeting mouse genes were cloned into LRC2.1T (U6-sgRNA-Cherry, Addgene# 108099). Single sgRNAs were cloned by annealing two DNA oligos and T4 DNA ligation into a BsmB1-digested LRC2.1T[69]. To improve U6 promoter transcription efficiency, an additional 5′ G nucleotide was added to all sgRNA oligo designs that did not already start with a 5′ G. gRNA sequences used to target *Slc2a6* are: CACCGAGCTGACTCCGAGGTCCAT and CACCGCACGGATGTGTTGTC GAAGA. gRNA sequence against the *ROSA* locus was used as a negative control: GAAGATGGGCGGGAGTCTTC.

**Protein purification and fluorescence polarization**. eIF4AI was expressed in *E. coli* BL21 cells and purified by Ni-NTA agarose and Q-Sepharose chromatography. Fluorescence polarization was performed by incubating 1 μM eIF4AI in fluorescence polarization buffer (14.4 mM HEPES-NaOH, 108 mM NaCl, 1 mM MgCl₂, 1 mM DTT, 14.4% glycerol and 0.1% DMSO, pH 7.5) in presence of either 1 mM ATP or AMP-PNP, 10 nM 5′-FAM-labelled (AG)₁₀ RNA and 50 μM CR-31 for 30 min at 25 °C. After incubation, the FAM-labelled RNA was chased with 100 μM unlabelled poly (AG)₁₀ RNA. For the combination of ATP and DMSO, 50 μM of eIF4AI was used instead due to low binding affinity of eIF4A to RNA. Variation in polarization was determined on a BMG Labtech Pherastar FS and the $t_{1/2}$ half-life calculated using one-phase exponential decay equation with Graphpad Prism software.

**Membrane fractionation**. Cytoplasmic and membrane protein extracts were prepared using the Subcellular Protein Fractionation Kit (Thermo Scientific Cat# 78840). In brief, cytoplasmic extraction buffer provided by the manufacturer was added to the cell pellet, followed by 15 min of incubation at 4 °C and centrifugation at 500 × g for 5 min. The supernatant containing the cytoplasmic protein extract was collected before the addition of membrane extraction buffer provided by the manufacturer, followed by 15 min of incubation at 4 °C and centrifugation at 3000 × g for 5 min. Supernatant containing the membrane protein extract was collected.

**NADP⁺/NADPH quantification**. NADP⁺/NADPH ratio was quantified by a luciferase assay provided in the NADP⁺/NADPH Glo Assay kit (Promega Cat# G9071). Patient-derived pancreatic cancer cells or murine organoids were plated in their respective media, incubated in a tissue culture incubator (37 °C, 5% CO₂), and were treated with CR-31, CB839 or BSO. After appropriate treatment over the desired time, the medium was removed and cells were supplemented with 1:1 PBS and 0.2 N NaOH solution with 1% DTAB to obtain a cell lysate. To measure NADP⁺, cell lysates were treated with 0.4 N HCl and heat quenched at 60 °C for 15 min. The solution was then neutralized with Trizma buffer (Tris base [Fisher Cat# BP152] in water). To measure NADPH, cell lysates were heat quenched at 60 °C for 15 min and the solution was neutralized with HCL-Trizma (Quality Biological Cat# 351-050-101). An equal volume of NADP⁺/NADPH-Glo Detection Reagent was added to each well with cell lysate, incubated at room temperature for 30 min, and luminescence was read on a luminometer (Spectramax i3x, Molecular Devices).

**Glucose Uptake Glo**. Glucose Uptake was measured using the Glucose Uptake-Glo Assay kit (Promega Cat# J1341). Cells were incubated in a tissue culture incubator (37 °C, 5% CO₂), and were treated with CR-31. After 5 h of treatment, the medium was replaced with PBS, and treated with 1 mM 2-deoxyglucose for 10 min. Stop Buffer, Neutralization Buffer, and 2DG6P Detection Reagent provided by the assay kit were subsequently added, incubated at room temperature for 30 min. Luminescence was read on a luminometer (Spectramax i3x, Molecular Devices).

**Metabolomics: Cell treatment and harvest**.

1. Organoids: One million cells were grown as organoids in ten 50 μl Matrigel domes in a 10 cm plate, cultured in EGF and NAC-free complete media and harvested at steady state or after treatment with DMSO or 100 nM CR-31 for 5 h. Spent media was collected at the end of the 5 h for metabolite analysis. For tracing experiments, ¹³C₆-Glucose (17.5 mM) or ¹³C₅-Glutamine (2 mM)-containing media (SITA media) was used to replace ¹²C media in the final 2 h. Organoids were collected on ice in 2 ml Cell Recovery Solution (Corning Cat# 354253) to remove Matrigel, triturated through a fire-polished glass pipette, and then washed with ice cold saline (9 g NaCl per litre). All saline was aspirated and discarded. The resultant pellet was immediately quenched with 1 ml −20 °C pre-chilled 80% methanol (v/v) and flash frozen on dry ice and stored at −80 °C until the day before GC/MS analysis.

2. Monolayer cells: One million patient-derived PDA cells were grown in DMEM supplemented with 10% dialyzed FBS (Gibco Cat# 26400044) and 1% PS, and treated with DMSO or 100 nM CR-31 for 5 h. For tracing experiments, ¹³C₆-Glucose (17.5 mM) or ¹³C₅-Glutamine (2 mM)-containing media (SITA media) was used to replace ¹²C media in the final 2 h. Spent media was collected at the end of the 5 h for metabolite analysis. Cells were rinsed with 10 ml of ice-cold saline (9 g NaCl per litre). All saline was aspirated and discarded. Cells were immediately quenched with 1 ml −20 °C pre-chilled 80% methanol (v/v), scraped and flash frozen on dry ice and stored at −80 °C until the day before GC/MS analysis.

**Metabolomics: stable isotope tracer analysis**. Sample extraction and GC/MS analysis were carried out as follows. Samples were subjected to sonication using BioRuptor (Diagenode Cat# UCD-200 TM,) for 10 min (30 s ON, 30 s OFF) at the "high" setting. Cellular debris was removed by centrifugation (10 min, 19,500 × g, 1 °C). Supernatants were transferred to fresh pre-chilled tubes containing 1 ml of 750 ng/ml D₂₇-myristic acid internal standard. Samples were subsequently dried overnight at −4 °C using a temperature controlled vacuum centrifuge (Labconco, Kansas City, MO, USA).

In order to protect and stabilize α-keto acids (pyruvate, α-ketoglutarate etc.), dried sample pellets were resuspended in 30 μl pyridine containing 10 mg ml⁻¹ methoxyamine hydrochloride (Sigma Cat# 226904), vortexed, and sonicated to ensure dissolution. Following 30 min of incubation at room temperature, samples were transferred to sealed autoinjection vials containing 70 μl of N-tert-Butyldimethylsilyl-N-methyltrifluoroacetamide (MTBSTFA) (Sigma Cat# M-108). Vials were incubated for 1 h at 70 °C.

Derivatized samples were subjected to GC/MS analysis using an Agilent 5975 C GC/MS equipped with a DB-5MS + DG (30 m × 250 μm × 0.25 μm) capillary column (Agilent J&W, Santa Clara, CA, USA). All samples were injected three times: twice using scan (50–700 m/z) mode (1× and 6× dilution) and once using selected ion monitoring (SIM) mode. One microlitre of the derivatized sample was injected in the GC in splitless mode with inlet temperature set to 280 °C and electron impact set at 70 eV. Helium was used as the carrier gas with a flow rate such that myristic acid eluted at approximately 18 min. The quadrupole was set at 150 °C and the GC/MS interface at 285 °C. The oven program started at 60 °C held for 1 min, then increasing at a rate of 10 °C min⁻¹ until 320 °C. Bake-out was at 320 °C for 10 min. All metabolites used in this study were previously validated using authentic standards to confirm mass spectra and retention times. Integration of ion intensities (generally M-57 ion) was done using the Agilent Mass Hunter Quant software (Agilent Technologies). Mass isotopomer distribution analysis was carried out using an in-house algorithm adapted from ref. [70].

**Metabolomics: extracellular analysis**. Spent media were collected from cultured cells and clarified by centrifugation for 10 min at 1 °C at 19,500 × g. Supernatants were submitted to the BioProfile 400 analyzer (Nova Biomedical Canada, Ltd. Mississauga, Ontario) for quantification of extracellular glucose and lactate.

**Extracellular acidification rates measurement (ECAR)**. Ten thousand cells were plated as organoids in 2 μl Matrigel domes or directly in monolayer in XF96 cell culture microplates (Seahorse Biosciences, Cat#101104-004) in EGF and NAC-free organoid complete media (for organoids) or DMEM supplemented with 10% FBS (for monolayer cells). After 24 h, growth media was changed to bicarbonate-free assay media (XF assay medium, Seahorse Biosciences, Cat#102365-100) and incubated at 37 °C for 1 h in a CO₂-free incubator. Extracellular acidification rate (ECAR) was measured using an XF96 Extracellular Flux Analyzer (Seahorse Biosciences) under basal conditions and following addition of glucose (10 mM), Oligomycin (1 μM), and

the glucose analog, 2-deoxyglucose, 2DG (50 mM), according to the manufacturer's protocol. Treatment was with vehicle or CR-31 for 5 h.

**Oxygen consumption rates measurement (OCR).** Ten thousand cells were plated as organoids in 2 µl Matrigel domes or directly in monolayer in XF96 cell culture microplates (Seahorse Biosciences, Cat#101104-004) in DMEM supplemented with 10% dialyzed FBS (for monolayer cells) or EGF and NAC-free organoid complete media (for organoids). After 24 h, growth media was changed to bicarbonate-free assay media (XF assay medium, Seahorse Biosciences, Cat#102365-100) and incubated at 37 °C for 1 h in a $CO_2$-free incubator. Oxygen consumption rate (OCR) was measured using an XF96 Extracellular Flux Analyzer (Seahorse Biosciences) under basal conditions or following the addition of Oligomycin (1 µM), the uncoupler FCCP (0.5 µM), and the electron transport inhibitor Rotenone/Antimycin A (0.5 µM), according to the manufacturer's protocol. Treatment was with vehicle or CR-31 for 5 h.

**Quantitative RT-PCR.** RNA was extracted from cell cultures using TRIzol reagent (Invitrogen Cat# 15596018). cDNA was synthesized using 1 µg of total RNA and TaqMan Reverse Transcription Reagents (Applied Biosystems Cat# N8080234). All targets were amplified (40 cycles) using gene-specific Taqman primers and probe sets (Applied Biosystems) on a QuantStudio5 Real-time-PCR instrument (Applied Biosystems). Relative gene expression quantification was performed using the ΔΔCT method with the QuantStudio Real-Time PCR software v1.1 (Applied Biosystems). Expression levels were normalized to *Hprt* and *Actb*.

Taqman probes used:
*HPRT* Mm00446968_m1, Hs02800695_m1
*ACTB* Hs99999903_m1
*NQO1* Hs01045994_m1
*HMOX1* Hs01110250_m1
*ENOLASE1* Hs00361415_m1
*HEXOKINASE II* Hs00606086_m1
*LDHA* Hs01378790_g1
*SLC2A1* Hs00214042_m1
*Slc2a6* Mm00554217_m1

**Western blot analysis.** Standard techniques were employed for immunoblotting of organoids. Organoids were quickly harvested using cold PBS on ice. Protein lysates were prepared using 0.1% SDS lysis buffer in 50 mM Tris pH8, 0.5% Deoxycholate, 150 mM NaCl, 2 mM EDTA, 1% NP40, with one tablet of Phos-STOP (Roche Cat# 4906837001) and one tablet of cOmplete™, Mini, EDTA-free Protease Inhibitor Cocktail (Roche Cat# 11836170001) per 10 ml buffer, and separated on 4–12% Bis-Tris NuPAGE gels (Invitrogen Cat# NP0335BOX), transferred onto a PVDF membrane (Millipore Cat# IPVH00010) and incubated with the following antibodies: TUBULIN (Cell Signalling Technology Cat# 2148, RRID:AB_2288042); ACTIN (Cell Signalling Technology Cat# 8456, RRID:AB_10998774), phospho-AKT (Cell Signalling Technology Cat# 4060, RRID:AB_2315049), pan-AKT (Cell Signalling Technology Cat# 9272, RRID:AB_329827), phospho-S6 Ribosomal Protein (Cell Signalling Technology Cat# 4858, RRID:AB_916156), S6 Ribosomal Protein (Cell Signalling Technology Cat# 2317, RRID:AB_2238583), 4EBP1 (Cell Signalling Technology Cat# 9452, RRID:AB_331692), phospho-4EBP1 (Cell Signalling Technology Cat# 3929, RRID:AB_10695878), eIF4E (Cell Signalling Technology Cat# 2067, RRID:AB_2097675), phospho-eIF4E (Cell Signalling Technology Cat# 9741, RRID:AB_331677), SLC2A6 (1:500) (Sigma-Aldrich Cat# HPA042272, RRID:AB_2677924), ENOLASE 1 (Cell Signalling Technology Cat# 3810, RRID:AB_2246524), ENOLASE 2 (Cell Signalling Technology Cat# 9536, RRID:AB_2099308), ENOLASE 3 (Abcam Cat# ab96334, RRID:AB_10680754), eIF4A1 (Abcam Cat#31217, RRID:AB_732122), eEF2 (Abcam Cat# ab75748, RRID:AB_1310165), IGF-IR-Iβ (Cell Signalling Technology Cat# 9750, RRID:AB_10950969), SDHB (Abcam Cat # ab14714, RRID:AB_301432), GAPDH (1:10,000) (Cell Signalling Technology Cat# 2118, RRID:AB_561053), VINCULIN (Cell Signalling Technology Cat# 4650, RRID:AB_10559207), eIF2α (Cell Signalling Technology Cat# 5324, RRID:AB_10692650), phospho-eIF2α (Cell Signalling Technology Cat# 3398, RRID:AB_2096481), SLC2A3 (Santa Cruz Biotechnology Cat# sc-74399, RRID:AB_1124975), SLC2A4 (Cell Signalling Technology Cat# 2213, RRID:AB_823508), HEXOKINASE I (Cell Signalling Technology Cat# 2024, RRID:AB_2116996), HEXOKINASE II (Cell Signalling Technology Cat# 2867, RRID:AB_2232946), LDHA (Cell Signalling Technology Cat# 3582, RRID:AB_2066887), mTOR (Cell Signalling Technology Cat# 2983, RRID:AB_2105622), phospho-mTOR (Cell Signalling Technology Cat# 5536, RRID:AB_10691552), PFKP (Cell Signalling Technology Cat# 8164, RRID:AB_2713957), PKM2 (Cell Signalling Technology Cat# 4053, RRID:AB_1904096), Pyruvate Dehydrogenase (Cell Signalling Technology Cat# 3205, RRID:AB_2162926), SLC2A1 (Cell Signalling Technology Cat# 12939, RRID:AB_2687899), Glutaminase (Abcam Cat# ab93434, RRID:AB_10561964 and Proteintech Group Cat#12855-1-AP, RRID:AB_2110381), KRAS (Abcam Cat# ab180772), AMPK (Cell Signalling Technology Cat# 2793, RRID:AB_915794), phospho-AMPK (Cell Signalling Technology Cat# 2535, RRID:AB_331250), affinity-purified rabbit-anti-full-length mouse NRF2 antibody (1:500) (made by JR Prigge, Montana State University) was a provided by Dr. Ed. Schmidt (Montana State University). All immunoblots are representative of at least three experiments. All antibodies were used at 1:1000 dilution unless stated otherwise.

**Immunohistochemistry.** Tissues were fixed in 10% neutral buffered formalin and embedded in paraffin. Sections were subjected to H&E staining as well as immunohistochemical staining. Antigen retrieval was done in 10 mM citrate buffer (pH 6). The following primary antibodies were used for immunohistochemical staining at the indicated dilution: phospho-Histone H3, 1:200 (Cell Signalling Technology Cat# 9701, RRID:AB_331535), c-MYC, 1:400 (Abcam Cat# ab32072, RRID:AB_731658), cleaved caspase 3, 1:200 (Cell Signalling Technology Cat# 9664, RRID:AB_2070042).

**Tumour bioluminescent and FDG-PET imaging analysis.** Athymic nude mice (Charles River) were orthotopically transplanted with luciferase-expressing pancreatic cancer cells. Upon detection of a mass by palpation, mice were subjected to bioluminescent imaging using the IVIS spectrum (PerkinElmer Cat# 124262). Mice were injected with 150 mg kg$^{-1}$ D-Luciferin (Goldbio Cat# LUCK) to ensure similar tumour volume (determined by total flux). Mice were randomized and enrolled one day after scanning. CR-31 was formulated in 5.2% PEG-400/5.2% Tween80 in saline. Mice were imaged once using FDG-PET, followed by treatment with vehicle or 0.2 mg kg$^{-1}$ CR-31. Sixteen hours after the first image, mice were given a second dose of vehicle or 0.2 mg kg$^{-1}$ CR-31 and imaged again using FDG-PET 5 h after.

FDG-PET: 6 h prior to the administration of FDG, animals were deprived of food with water source remaining. [$^{18}$F]-FDG was administered by intraperitoneal injection in a dose of 3.7 MBq per mouse. Approximately 60 min after FDG administration, static PET images of the whole body were acquired over 20 min using Inveon D-PET scanner (Siemens) in 2D mode. Immediately after completion of the pre-treatment/baseline PET scan, treatment was administered to the animals, as described above. Food was returned to the animals at that point.

Analysis: The images were reconstructed using Inveon software in 2D OSEM mode with 4 iterations and 128 × 128 matrix. Image analysis was performed on PBAS software (Pmod Technologies LLC, Zurich, Switzerland). The volumes of interest were drawn around the tumours in the right side of the abdomen and around the contralateral thigh muscles. SUVmax values were used for quantitative analysis. In order to compensate for the metabolic variabilities between the animals, tumour to muscle ratio was also calculated.

**Reporting summary.** Further information on research design is available in the Nature Research Reporting Summary linked to this article.

## Data availability
The polysome profile datasets that support the findings of this study have been deposited in GEO under the accession number **GSE125380**. In addition to polysome profiling datasets, all other raw data that support the findings of this study are also provided in "Source Data file".

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

## Acknowledgements

We thank Dr. William Devine (Boston University) for the synthesis of CR-31, members of the Chio lab for critical discussion, Dr. Richard Baer (Columbia University) for critical review of the manuscript, Dr. Edward Schmidt (Montana State University) for the generous gift of the anti-murine NRF2 antibody, and Dr. David Tuveson (Cold Spring

Harbor Laboratory) for sharing patient-derived pancreatic cancer cells. This work was performed with the support of Herbert Irving Comprehensive Cancer Center (Columbia University) Flow Cytometry, Small Animal Imaging, and Molecular Pathology Shared Resources, which are supported by the Cancer Center Support Grant P30CA013696. Sequencing was performed at the McGill University and Génome Québec Innovation Centre, Montreal, Canada. NOVA, GC/MS, and SITA analyses were performed by the GCRC Metabolomics Core Facility funded by the Dr. John R and Clara M. Fraser Memorial Trust, the Terry Fox Foundation, the Québec Breast Cancer Foundation, and McGill University. FDG-PET studies were done with the support of Columbia University Department of Radiology and NCATS UL1TR001873 (Reilly) Irving Institute/CTSA Translational Therapeutics Accelerator. I.I.C.C. is supported by the Pancreatic Cancer Action Network (PG009667 - PANCAN 18-35-CHIO); the V Foundation (PG009685 - VFND V2018-017); and Columbia University Medical Center (Paul Marks Scholar Award). J. Pelletier and M.P. are supported by the Canadian Institutes of Health Research (CIHR) (J. Pelletier (FDN-148366)) and the Canadian Cancer Research Institute (J. Pelletier and M.P. (705096)). O.L. is supported by the Swedish Research Council, Swedish Cancer Society, Stockholm Cancer Society, and the Wallenberg Academy Fellow program. J.A.P., Jr. is supported by the National Institutes of Health (NIH) (R35 GM118173). Work at the BU-CMD (J.A.P., Jr. and L.E.B.) is supported by the NIH (R24 GM111625).

## Author contributions

K.C. performed and analyzed the following experiments: Viability assays, immunoblot analyses of signalling pathways, lysate fractionation, qPCR, heavy isotope tracing of glucose and glutamine, GSH/GSSG and NADP$^+$/NADPH quantification, DCFDA flow cytometry, seahorse experiments, glucose uptake assays, CRISPR-mediated silencing of *Slc2a6* in PDA cells, viral transduction of NDI1, drug sensitivity in vitro and in vivo on orthotopic transplant mice and C57Bl/6J mice. F.R. performed polysome profiling, sequencing and subsequent RT-qPCR validations on organoids. F.R. cloned hairpins against eIF4A. C.O. performed analysis of all data pertaining to polysome profiling. D.K. L. and M.D. performed and analyzed bioluminescence and FDG PET imaging experiments. D.K.L., J. Park, J.G., and F.G. performed immunostaining and scoring of phospho-Histone H3, cleaved caspase 3, and c-MYC. B.D., C.S., M.Y., A.H.S., and I.I.C.C. performed in vivo animal study on KPC mice. I.I.C.C. isolated murine organoid lines, perform orthotopic transplantations and performed experiments to measure mRNA translation. M.P. assisted with the analysis of all metabolic experiments. J.S. designed sgRNAs against *Slc2a6*. D.A. ran and analyzed GC/MS experiments. C.J.T.
provided CB839 and assisted with analysis of metabolic experiments. J.A.P., Jr. and L.B. provided CR-31. J. Pelletier, O.L., and I.I.C.C. designed the study and wrote the manuscript.

## Competing interests

The authors declare no competing interests.
