## [Peer Review File · Nature Communications]

Reviewers' comments:

Reviewer #1 (Remarks to the Author):

This is an interesting manuscript that reports on the rocaglate CR-31 and its effects on cancerous cells. In contrast to a previous report from the Ingolia group that reported that RocA increased the clamping of eIF4A on mRNAs in polypurine tracts in the 5' UTR, CR-31 inhibited mRNAs that either had shorter 5' UTRs or had elevated GC content. Additionally, a different observation was made on the mRNAs most influenced by CR-31, mRNAs associated with proteins in "central carbon metabolism" and the electron transport chain.

A second interesting finding was that the metabolism of glutamine was perturbed. CR-31 reversed the normal response in the uptake of glutamine and increase in TCA cycle intermediates that generally occurs in cancerous cells and inhibited this conversion through the reduction in the synthesis of the GLS1 protein and the transporter Slc1□5. Consistent with this observation was the enhanced tumor suppression activity of CR-31 combined with an inhibitor of glutaminase, CB839. What is left open is the extent to which such treatment might also be useful in defeating other tumors.

Specific concerns

1. Introduction – "Recoglates cause eIF4A to clamp ... therefore thought to restrict efficient recycling of eIF4A..." The suggestion from the Ingolia group was that RocA inhibited mRNA translation due to the clamped eIF4A on specific mRNAs. Given the 10-fold excess of eIF4A over eIF4G, it is unlikely that there is insufficient eIF4A for eIF4G recycling.
2. Figure S2 – for the normal cells treated with 100 nM CR-31, there appears to be an increase in p-mTOR, p-eIF4E and p-4EBP1 which would normally be consistent with enhanced protein synthesis although this effect does not appear to be very evident in KP1 or KP2. The text says "...we did not observe a decrease in the activation status". Secondly, for the KP1 and KP2 cells, there appears to be an enhanced level of p-4EBP1, yet there is no comment. In a similar manner, in panel d, for the KP cells it appears that 100 nM CR-31 leads to a decrease in eIF2□ phosphorylation. This begs the question as to how reproducible these findings are and what their implication might be. In addition, the text indicates "...it only did so at concentrations above the IC50 range for PDA cells." Figure S1 indicates the IC50 values for N as 48 nM and for KP as 5.2 nM. However, the loss of cMyc is only seen at 100 nM, not 10 nM for the KP cells. Is there an explanation?
3. In keeping with ribosomal profiling, the authors have accumulated a considerable amount of data. However, it is not clear that all the interpretations from this data have been made. For

example, the data in Figure 3 provide a clear explanation for the up/down regulation for the expression of members of the electron transport chain. Are there other systems for which an equivalent figure might be prepared which would give the reader a better idea of the systematic effects of CR-31 (i.e. glycolysis, fatty acid synthesis, etc.).

4. The subsequent analysis of protein levels is important as a reflection of both protein synthesis as well as protein turnover. In this light, one would anticipate greater differences for proteins of shorter half life. Do the authors have estimates of the half lives for the proteins analyzed in Figure S5? Second, in Figure S6 panel c, CR-31 treatment at 100 nM appears to lead to an increase in Kras yet the comment in the text is “ ... there was no observable decrease in KRAS expression level...”.

5. The designation of M+3, M+4 and M+5 are now well explained in the figure legends (Figure 4 and 5) or text and this needs to be improved.

6. The authors conclude that “eIF4A clamping cannot explain the effect of CR-31 ...”. The best proof of this would be the in vitro binding experiment that shows there is no difference in the off rate of eIF4A from an RNA complex formed in the presence of ATP (+/- CR-31). This must be done.

Minor concerns

1. In general, eIF4A is required for the translation of most mRNAs. This reviewer feels that the start of the title is misleading (eIF4A-dependent translational control) as all mRNAs could be influenced by levels of active eIF4A. An alternate such as “Inhibition of eIF4A by CR-31 influences control of central carbon metabolism...” might be more appropriate.

2. The authors define the abbreviation of CR-1-31-B as CR-31. Then why have they not systematically used this abbreviation in their manuscript (see text and figures)? They should. In a similar manner, the authors use h, hr, hrs and hours in the manuscript. They should use the journal approved abbreviations for time. There appears to be a mix of title types used in the references (some journal names are abbreviated, some are not). In addition, some have “doi” designations, others do not. The authors need to use the uniform referencing used by the journal.

3. Figure 5 d – it would be good to include glutamic acid in between glutamine and \square ketoglutarate as the conversion of glutamine requires both glutaminase and glutamate dehydrogenase.

4. Table S2 – CR-31 treatment leads to the upregulation of 1061 mRNAs and the down regulation of 1169 mRNAs. Which is more important, the upregulation or the down regulation (and why)?

Reviewer #2 (Remarks to the Author):

In the manuscript by Chan et al, the authors showed that pharmacological inhibition of eIF4A with CR-31 specifically target pancreatic cancer cells at low dose. CR-31 treatment not only induces PDA cell death in organoid culture, but also suppresses tumor growth in vivo and significantly prolongs animal survival in KPC model. At molecular level, the authors showed that low dose CR-31 selectively suppresses the translation of mRNAs with highly structured 5' UTR. As consequence, CR-31 treatment suppresses both mitochondria OXPHOS and glycolysis which is correlated with the translational inhibition of several genes involved in those metabolism pathways. In addition, the authors also observed induction of reductive carboxylation, elevated ROS and increased translation of genes involved in glutamine uptake and utilization. Targeting glutamine metabolism further sensitizes tumor cells to CR-31. Overall, the manuscript identified targeting eIF4A as a novel approach to target PDA and offered interesting molecular insight on the selective-dependency of eIF4A-mediated translation in tumor cells. Additional control experiments are needed to further address the tumor-specific sensitivity of CR-31. Here're the major concerns.

1. The authors nicely showed that CR-31 selectively kills tumor cells at low dose. What's the effect of genetic depletion of eIF4A in KP and N organoid?
2. Do the authors know the circulating CR-31 level in the in vivo studies? Is the dose used in the in vivo models reflective of the low dose that selectively kills the tumor cells in vitro?
3. While the authors showed that the CR-31-sensitive mRNAs contain relatively structured 5'UTR, it's not clear why the translation of these genes are selectively suppressed in KP organoid compared to N organoid. Is the 5'UTR different or is the loading of eIF4A to 5'UTR different between the organoids?
4. The authored showed that CR-31 inhibits the translation of several genes involved in OXPHOS and glucose metabolism. Are the basal transcription or protein levels of these genes different between KP and N organoid? Is the basal OXPHOS and glycolysis different between KP and N organoid? Does CR-31 also suppress the OXPHOS and glycolysis in N organoid?
5. It will be better to further clarify the effect of CR-31 on glycolysis using ¹³C-glucose tracing experiment. In addition, what does the ECAR data look like if the authors do baseline normalization

in Fig.4e. Of note, the dose of CR-31 used in Fig.4d is 100nM, in contrast to the 10nM used in other experiments. Please confirm if it's correct or a typo.

6. The authors tried to claim that the downregulation of Slc2a6 is likely accounted for the decrease in glucose uptake upon CR-31 treatment. As the authors pointed out in the discussion, Slc2a1 and Slc2a3 are the major glucose transporters expressed in tumor cells. A lot more evidence is needed if the authors want to claim that the suppression of Slc2a6 is indeed the mechanism for the decrease of glucose uptake upon CR-31 treatment (please see my general comment in the end). Is Slc2a6 protein level in general higher in tumor cells compare to normal cells? Does Slc2a6 depletion suppress glucose uptake and whether reconstitution of Slc2a6 expression in CR-31 treated cells restores glucose uptake?

7. The authors showed that GLS1 inhibition sensitizes PDA cells to CR-31. What's the impact of GLSi alone or in combination with CR-31 on NADPH/NADP ratio and cellular ROS level? Does other agents affect NADPH metabolism or ROS regulation, such as BSO, also sensitize cells to CR-31? Will the cytotoxicity of CR-31 or the CR-31/CB839 combination be rescued with ROS scavenger?

8. In general, the authors provided a comprehensive characterization of the impact of CR-31 treatment on cellular metabolism. In addition, the authors also observed changes in the translation of multiple genes in those metabolism pathways, such as ETC, glucose uptake and glutamine metabolism. It's important for the authors to present these data to readers. However, mechanistic speculations should be avoided. For example, is the downregulation of Slc2a6 by CR-31 potent enough to suppress glucose uptake in PDA cells? Does the induction of reductive carboxylation have anything to do with the change in NADPH/NADP ratio and ROS level upon CR-31 treatment? In my opinion, the observations are interesting enough by themselves although the data are mostly correlative at current state. There's no need for the authors to offer mechanistic explanations to every observation in one manuscript. But the authors should consider re-organizing the data description and clarifying the discussion to avoid potentially misleading the readers.

Reviewer #3 (Remarks to the Author):

Chan, Robert, Oertlin, Kapeller-Libermann, and colleagues study the role of eIF4A-mediated translation in regulating the metabolic pathways of human and murine pancreatic cancer cells. The study uses multiple in vitro systems with some in vivo validation to argue that inhibition of eIF4A-controlled translation by CR-31 specifically alters the metabolic state of tumor cells resulting in increased oxidative stress and reliance on glutamine for compensation in the synthetic and redox pathways.

The idea is conceptually interesting and the experiments provide some proof-of-principle support. I would ask for the following issues to be addressed:

1. The authors use normal ducts and duct-derived organoids as their comparison control group. This comparison is used to argue for CR-31's efficacy and to identify the changes in polysome-associated RNAs. Given that the KP organoids are actually acinar cell-derived in these mice, this control makes less sense to me. Why were these studies performed that way and do the authors have any data from native acinar cells? If not, could the authors at least replicate the viability and protein synthesis control experiments with acinar cell controls.
2. Was there any toxicity seen in the mice during the tumor treatments? Were mouse weights recorded during these experiments and could this data be provided?
3. The statements revolving around CR-31's selectivity specifically for pancreatic cancer cells are too strong in this manuscript. The authors do not include nearly enough controls to really prove that. I would tone down the language. Example of this can be found on page of the Results – start of the mRNA modulation section.
4. Could the authors clarify in their Methods or Results somewhere what they mean by “four pairs of murine N and KP organoids”? Was this a true paired design, where “normal” pancreas was obtained from KP mice or is this simply four normal mice and four KP mice?
5. The section on 5'UTR region composition and structure is interesting although a bit ancillary to the rest of the manuscript. I would caution the authors about the statement made regarding similarity to the general structure features found in the Rubio manuscript. In that paper, the silvestrol-regulated transcripts with lower translation had longer 5'UTRs and lower GC content. This is opposite to what is seen in the CR-31 sensitive transcripts, which have shorter 5'UTR with a higher GC content.
6. I generally like the experimental work in the carbon metabolism section. I think the data regarding the ETC inhibition and oxygen consumption are solid. I would again caution the authors regarding their language for the results of the shRNA eIF4A experiment. This experiment phenocopies the CR-31 data (as the authors rightly point out in their Discussion). It is not a proof of CR-31's mechanism specificity through eIF4a inhibition.
7. The glycolysis inhibition data is less consistent. The extracellular acidification rate is certainly reduced. However, the protein level data does not suggest any significant changes of enzyme levels.

The steady state labeling with ^{13}C glucose results are also hard to interpret. The data shows a decrease in labeling of glycolytic intermediates, but this is just 5% change in most of the intermediates measured here. It is difficult to square that with the acidification and lactate level data unless we know how long the ^{13}C glucose was in the system given that glycolytic flux is a fast process in pancreatic tumor cells. Could the authors provide information on how long they exposed the cells to ^{13}C glucose and more specifically discuss other carbon sources in their system in their Methods or Results?

8. As a sidenote, the way of displaying the data of the stable metabolic labeling is at best unintentionally misleading with the choice of Y axis ranges to maximize and magnify the apparent difference between the treatments. If the axes were ranged from 0, the small magnitude of the changes would be more faithfully demonstrated.

My overall take from the paper is that regulation of translation certainly affects tumor cell metabolism. The data to support this as a possible therapeutic approach are much less clear (partly due to the choice of the control using “normal ducts”) and the authors should temper their comments here. I found the experimental results interesting by themselves without having to resort to the “sales pitch” of treatment. The glycolysis experiments need to be clarified. Thank you for allowing me to review this work and I look forward to the revision!

We thank the reviewers for their constructive comments and suggestions. After addressing the raised issues, our revised manuscript more clearly demonstrates the role of cap-dependent translation initiation in supporting central carbon metabolism in pancreatic cancer. We have further extended our analyses to characterize the impact of eIF4A inhibition on PDA redox homeostasis. We provide additional evidence that the glucose transporter SLC2A6 is functional in pancreatic cancer cells to support glucose uptake and glycolysis and that its translation is sensitive to pharmacological inhibition of eIF4A using the synthetic rocaglate, CR-31. CR-31-treated pancreatic cancer cells exhibit increased dependency on glutamine reductive carboxylation as the major pathway of citrate formation. We now provide additional evidence that combined inhibition of eIF4A and glutaminase more effectively inhibited the growth of pancreatic cancer cells in vitro and in vivo. We hope that our improved manuscript is deemed suitable for publication in Nature Communications. To include the new data and for clarity, we have moved some data into different figures or supplemental figures. The datasets generated during and/or analysed during the current study are available in the GEO repository under the accession number GSE125380. Data can be previewed using the following access token: opozcaiczbizdul.

Reviewer #1 (Remarks to the Author):

This is an interesting manuscript that reports on the rocaglate CR-31 and its effects on cancerous cells. In contrast to a previous report from the Ingolia group that reported that RocA increased the clamping of eIF4A on mRNAs in polypurine tracts in the 5' UTR, CR-31 inhibited mRNAs that either had shorter 5' UTRs or had elevated GC content. Additionally, a different observation was made on the mRNAs most influenced by CR-31, mRNAs associated with proteins in “central carbon metabolism” and the electron transport chain.

A second interesting finding was that the metabolism of glutamine was perturbed. CR-31 reversed the normal response in the uptake of glutamine and increase in TCA cycle intermediates that generally occurs in cancerous cells and inhibited this conversion through the reduction in the synthesis of the GLS1 protein and the transporter Slc1 α 5. Consistent with this observation was the enhanced tumour suppression activity of CR-31 combined with an inhibitor of glutaminase, CB839. What is left open is the extent to which such treatment might also be useful in defeating other tumours.

We appreciate the reviewer's recognition of the potential importance of our work, and have endeavoured to strengthen and clarify our initial findings in the revised manuscript.

Specific concerns

1. Introduction – “Recoglates cause eIF4A to clamp ... therefore thought to restrict efficient recycling of eIF4A...” The suggestion from the Ingolia group was that RocA inhibited mRNA translation due to the clamped eIF4A on specific mRNAs. Given the 10-fold excess of eIF4A over eIF4G, it is unlikely that there is insufficient eIF4A for eIF4G recycling.

Response: The reviewer is correct that work done by the Ingolia group previously demonstrated that RocA specifically clamps eIF4A onto polypurine sequences (RocA-eIF4A target sequence). This was proposed to block 43S scanning, leading to premature, upstream translation initiation and reduced protein expression from transcripts bearing the RocA-eIF4A target sequence. However, polypurine sequences are not unique to mRNAs and are also found in rRNAs. As such, clamping of eIF4A onto rRNA-ribosome complexes could in theory deplete free-eIF4A. Indeed, it has been shown using m⁷GTP affinity purification that one effect of silvestrol and (-)-9 (also known as CR-31) is depletion of eIF4A1/II from the eIF4F complex¹. Therefore, to improve clarity, we have now revised the discussion of the manuscript to read as follows: “While ours and previous studies²⁻⁴ support the notion that modulation of eIF4A activity leads to selective effects on translation, the underlying mechanisms are still incompletely understood. It has been suggested that G-quadruplexes in 5'

UTRs mediate selective suppression of translation downstream of eIF4A inhibition by the rocaglate silvestrol⁴, while eIF4A clamping to polypurine sequences underlie the effects of the rocaglate Rocaglamide A (RocA)². Our findings suggest that in PDA cells slightly shorter 5' UTRs with more stable secondary structures contribute to the selective suppression of translation following CR-31 treatment. Importantly, we observed that shRNA depletion of eIF4A phenocopies the metabolic changes induced by CR-31. In conjunction with our finding that polypurine sequences are not enriched in mRNAs translationally suppressed by CR-31, we conclude that eIF4A clamping on polypurine sequences cannot account for the full effect of CR-31 on translational changes observed herein."

2. Figure S2 – for the normal cells treated with 100 nM CR-31, there appears to be an increase in p-mTOR, p-eIF4E and p-4EBP1 which would normally be consistent with enhanced protein synthesis although this effect does not appear to be very evident in KP1 or KP2. The text says "...we did not observe a decrease in the activation status". Secondly, for the KP1 and KP2 cells, there appears to be an enhanced level of p-4EBP1, yet there is no comment. In a similar manner, in panel d, for the KP cells it appears that 100 nM CR-31 leads to a decrease in eIF2 α phosphorylation. This begs the question as to how reproducible these findings are and what their implication might be. In addition, the text indicates "...it only did so at concentrations above the IC50 range for PDA cells." Figure S1 indicates the IC50 values for N as 48 nM and for KP as 5.2 nM. However, the loss of cMyc is only seen at 100 nM, not 10 nM for the KP cells. Is there an explanation?

Response:

To address the issue of reproducibility, we have now extended our analysis to evaluate these signaling changes at additional timepoints to assess the effects of eIF4A inhibition upon CR-31 treatment. The same analysis was performed on patient-derived PDA cell lines. Data from all biological replicates are now quantified and presented graphically (**Supplementary Fig. 2d**). As shown in the revised Supplementary Figure 2, N organoids exhibit an induction of pS6 and a decrease in pAKT upon CR-31 treatment (**Supplementary Figs. 2a, 2d**). However, no change was observed for p4EBP1 (**Supplementary Figs. 2a, 2d**). These changes were not observed in KP organoids. In KP organoids, we noted a transient induction of eIF4E phosphorylation upon CR-31 treatment but these changes were no longer observed after 5 h (**Supplementary Figs. 2b, 2d**). Moreover, this transient induction was not observed in human PDA cells (**Supplementary Fig. 2c**). We therefore conclude that the inhibition of translation upon CR-31 treatment was not a consequence of indirect suppression of the PI3K/AKT/mTOR or MEK/MNK/eIF4E signaling pathways although there seems to be possible compensatory mechanisms upon acute suppression of translation initiation by CR-31. The significance of these compensatory mechanisms is part of our ongoing studies.

To address the role of c-MYC in the cytotoxicity of CR-31, we examined the expression level of c-MYC in vivo upon CR-31 treatment (**Supplementary Fig. 2f**). We confirmed that at a dosage which exhibits tumour suppressive effects in vivo with minimal systemic toxicity, CR-31 does not suppress c-MYC expression in tissues collected 2 h post CR-31 treatment. As described in the previous version of our manuscript, c-MYC is sensitive to 100 nM, but not 10 nM CR-31 in both N and KP

Appendix 1

a) Immunoblot analysis of c-MYC in N and KP organoids treated with CR-31. **b)** Immunoblot analysis of c-MYC in patient derived PDA cells treated with CR-31. **c)** Left, Glycolysis reflected by extracellular acidification rate (ECAR) was measured in primary human pancreatic cancer cells after 5 h of treatment with vehicle or the indicated concentrations of (+)JQ1. ECAR was measured before and following the addition of glucose (10 mM), Oligomycin (1 μ M) and the glucose analog, 2-deoxyglucose, 2DG (50 mM). Data are mean \pm S.D., n=5 technical replicates each. Right, Immunoblot analysis c-MYC expression upon vehicle or (+)JQ1 treatment. Vinculin, loading control.

organoids (**Appendix 1a, 1b**). The insensitivity of *c-MYC* mRNA to 10 nM CR-31 was also confirmed in our polysome profiling experiment. Although *c-MYC* expression can be suppressed by 100 nM CR-31, its target genes (*LDHA*, *ENO1*, *HK2* and *SLC2A1*) were not decreased in PDA cells treated with 10 nM or 100 nM of CR-31 (**Supplementary Figs. 7a, 7b**). Moreover, alternative means of *c-MYC* inhibition using agents such as the bromodomain inhibitor (+)JQ-1⁵ did not decrease glycolysis in PDA cells (**Appendix 1c**) at concentrations where *c-Myc* expression was effectively suppressed. The dose-dependent sensitivity of *c-MYC* to CR-31 is part of our ongoing investigation. Since *c-MYC* expression is not decreased in tumours treated with CR-31 and also does not seem to account for the glycolytic phenotype observed herein, we have excluded in depth description of this in the revised manuscript.

3. In keeping with ribosomal profiling, the authors have accumulated a considerable amount of data. However, it is not clear that all the interpretations from this data have been made. For example, the data in Figure 3 provide a clear explanation for the up/down regulation for the expression of members of the electron transport chain. Are there other systems for which an equivalent figure might be prepared which would give the reader a better idea of the systematic affects of CR-31 (i.e. glycolysis, fatty acid synthesis, etc.).

Response: We thank the reviewer for this suggestion. As recommended, we performed our analysis using a different pathway database (KEGG) to better demonstrate the systemic effects of CR-31. This data is now presented in our revised Supplementary Fig. 5. Here, we show that two metabolic pathways are translationally upregulated in KP vs N organoids and subsequently down regulated in CR-31 treated KP organoids: glutathione metabolism and oxidative phosphorylation. We have extended our analyses to characterize the redox consequences of CR-31 in PDA cells. These data are now presented in revised Fig. 3.

4. The subsequent analysis of protein levels is important as a reflection of both protein synthesis as well as protein turnover. In this light, one would anticipate greater differences for proteins of shorter half-life. Do the authors have estimates of the half-lives for the proteins analyzed in Figure S5?

Response: As recommended by the reviewer, we evaluated the protein stability of GAPDH, Enolase1/2/3, and SLC2A6. As shown in revised Supplementary Fig. 7h, SLC2A6 protein level is significantly reduced after 12 h of cycloheximide treatment whereas that of GAPDH and ENOLASE1/2/3 remain unchanged. This likely explains the stronger impact of CR-31 on SLC2A6 than on Enolase2, Enolase 3, and GAPDH even though the translation of all four mRNAs are reduced to similar extents by CR-31 (Fig. 4i). Indeed, after 24 h of CR-31 treatment, we begin to observe moderate decreases in GAPDH, ENOLASE2 and ENOLASE3 expression in both murine and patient-derived PDA cells (Fig. 4k, Supplementary Fig. 7g).

Second, in Figure S6 panel c, CR-31 treatment at 100 nM appears to lead to an increase in *Kras* yet the comment in the text is “... there was no observable decrease in *KRAS* expression level...”.

*Response: To address this issue, we have extended our analysis to evaluate the effect of CR-31 on *KRAS* expression level at additional timepoints in both murine organoids and patient-derived PDA cell lines. Data from all biological replicates analyzed are now quantified and presented graphically (Supplementary Figs. 7c-f). As shown in the revised figures, while there is a transient induction of *KRAS* at 1 h after CR-31 treatment, this subsided at the 5 h timepoint, which is the timepoint used for all our metabolism experiments. We have now clarified this observation in our revised manuscript. Collectively, our data indicate that the metabolic alterations occurring upon CR-31 treatment are independent of suppressed *KRAS* expression.*

5. The designation of M+3, M+4 and M+5 are now well explained in the figure legends (Figure 4 and 5) or text and this needs to be improved.

Response: To improve clarity, we have included a schematic to demonstrate the flow of carbons downstream of ¹³C-glucose (Fig. 4f) and ¹³C-glutamine (Fig. 5d) to better illustrate the designation of M+3, M+4 and M+5

as isotopologues of metabolic intermediates in these pathways. We have also revised the legend of **Fig. 4f** to read as follows: “Top, schematic illustrating the flow of heavy carbons through the glycolytic pathway, i.e., the M+3 labelled lactate and pyruvate are isotopologues of lactate and pyruvate with three labelled carbons derived directly from $^{13}\text{C}_6$ -glucose.”

6. The authors conclude that “eIF4A clamping cannot explain the effect of CR-31 ...”. The best proof of this would be the in vitro binding experiment that shows there is no difference in the off rate of eIF4A from an RNA complex formed in the presence of ATP (+/- CR-31). This must be done.

Response: We thank the reviewer for recommending this experiment. We have now performed the experiment and we observed that CR-31 does impact the off rate of eIF4A from an RNA complex (Supplementary Fig. 1d). However, in contrast to a previous study on another rocaglate, RocA², we did not observe any enrichment for polypurines in the 5' UTRs of transcripts that are translationally suppressed in KP organoids after CR-31 treatment (Supplementary Fig. 4a). We have revised our discussion to more accurately reflect these observations. Upon revision, our discussion reads as follows: “...we conclude that eIF4A clamping on polypurine sequences cannot account for the full effect of CR-31 on translational control observed herein.”

Minor concerns

1. In general, eIF4A is required for the translation of most mRNAs. This reviewer feels that the start of the title is misleading (eIF4A-dependent translational control) as all mRNAs could be influenced by levels of active eIF4A. An alternate such as “Inhibition of eIF4A by CR-31 influences control of central carbon metabolism...” might be more appropriate.

Response: We thank the reviewer for this suggestion. We have revised the title of our manuscript accordingly: “eIF4A supports an oncogenic translation program in pancreatic ductal adenocarcinoma”

2. The authors define the abbreviation of CR-1-31-B as CR-31. Then why have they not systematically used this abbreviation in their manuscript (see text and figures)? They should.

Response: We have now unified the abbreviation throughout the revised manuscript.

In a similar manner, the authors use h, hr, hrs and hours in the manuscript. They should use the journal approved abbreviations for time. There appears to be a mix of title types used in the references (some journal names are abbreviated, some are not). In addition, some have “doi” designations, others do not. The authors need to use the uniform referencing used by the journal.

Response: We have now unified these abbreviations throughout the revised manuscript in accordance to the referencing style of Nature Communications.

3. Figure 5 d – it would be good to include glutamic acid in between glutamine and α -ketoglutarate as the conversion of glutamine requires both glutaminase and glutamate dehydrogenase.

Response: We did not observe a change in the contribution of ^{13}C -glutamine to glutamic acid (**Appendix 2**) even though there is a decrease in the downstream oxidative (M+4) metabolites and an increase in the downstream reductive (M+5) metabolites of glutamine. This was observed in both patient-derived PDA cells (**Figs. 5d, e**) and KP organoids (**Fig. 5f**). These changes were not observed in N organoids (**Fig. 5f**), consistent with our observations that PDA cells exhibit heightened sensitivity to eIF4A inhibition when compared to normal counterparts. Since glutamic acid is channelled into both the TCA cycle and to glutathione synthesis, the levels of ^{13}C labelled glutamic acid will be influenced by the activities of GLS (glutamine to glutamate conversion), GDH (glutamate to α -ketoglutarate conversion) and also GCLC (synthesis of glutathione from glutamate, glycine and cysteine). We speculate that shorter timepoints of ^{13}C tracing would be required to capture the change in ^{13}C labelled glutamic acid levels.

Appendix 2

Steady state intracellular glutamine levels from 4 patient-derived pancreatic cancer cell lines treated with vehicle or 100 nM CR-31 for 5 h. Data are mean \pm S.D., Paired *t* test. $^{13}\text{C}_6$ -glutamine tracing of 4 patient-derived pancreatic cancer cell lines upon treatment with vehicle or 100 nM CR-31 for 5 h. Data are mean \pm S.D., Paired *t* test.

4. Table S2 – CR-31 treatment leads to the upregulation of 1061 mRNAs and the down regulation of 1169 mRNAs. Which is more important, the upregulation or the down regulation (and why)?

Response: Indeed, similar observations have been reported in prior studies but the underlying basis for translational upregulation of mRNAs upon eIF4A inhibition^{3,4,6,7} has not been explored. As presented in our manuscript, both the upregulated and the downregulated subsets of mRNA are functionally relevant. Translational downregulation of the glucose transporter SLC2A6 and the components of the electron transport chain reduced the level of glycolysis and mitochondrial oxidative phosphorylation, respectively. On the other hand, translation upregulation of the glutamine transporter (SLC1A5) and the enzyme, glutaminase (GLS), increased glutamine metabolism. These events were evident in PDA cells but not in normal proliferating counterparts and may in part reflect competition of mRNAs for translation factors. Nonetheless, we speculate that this may also reflect an active adaptive mechanism that the translation machinery engages in response to stress during pathological (or physiological) conditions. This is now elaborated in our revised discussion.

Reviewer #2 (Remarks to the Author):

In the manuscript by Chan et al, the authors showed that pharmacological inhibition of eIF4A with CR-31 specifically target pancreatic cancer cells at low dose. CR-31 treatment not only induces PDA cell death in organoid culture, but also suppresses tumour growth in vivo and significantly prolongs animal survival in KPC model. At molecular level, the authors showed that low dose CR-31 selectively suppresses the translation of mRNAs with highly structured 5' UTR. As consequence, CR-31 treatment suppresses both mitochondria OXPHOS and glycolysis which is correlated with the translational inhibition of several genes involved in those metabolism pathways. In addition, the authors also observed induction of reductive carboxylation, elevated ROS and increased translation of genes involved in glutamine uptake and utilization. Targeting glutamine metabolism further sensitizes tumour cells to CR-31. Overall, the manuscript identified targeting eIF4A as a novel approach to target PDA and offered interesting molecular insight on the selective-dependency of eIF4A-mediated translation in tumour cells. Additional control experiments are needed to further address the tumour-specific sensitivity of CR-31. Here're the major concerns.

1. The authors nicely showed that CR-31 selectively kills tumour cells at low dose. What's the effect of genetic depletion of eIF4A in KP and N organoid?

Response: Genetic ablation of eIF4A1 is embryonically lethal (ongoing, unpublished). We therefore speculate that genetic depletion of eIF4A will have effects in both KP and N organoids. Indeed, 72 hours of 100 nM CR-31 treatment led to cytotoxicity in both N and KP organoids (Fig. 1b). In summary, our data support that the sensitivity to eIF4A inhibition is dependent on mRNA selectivity and dosage, thus providing a therapeutic window to target eIF4A. Complete elimination of eIF4A will be detrimental to all cell types.

2. Do the authors know the circulating CR-31 level in the in vivo studies? Is the dose used in the in vivo models reflective of the low dose that selectively kills the tumour cells in vitro?

Response: Circulating levels of CR-31 is difficult to assess as 82-84% of CR-31 in plasma is bound to plasma proteins¹. In a previous study, mice receiving repeated daily injections at 0.2 mg kg⁻¹ CR-31 (same dose used in current study) exhibited no observable systemic toxicity and CR-31 did not display toxicity toward cells of hematopoietic origin¹. This is consistent with our new data showing that body weight, liver, pancreatic enzymes and gross histology are not changed in CR-31-treated mice compared to vehicle control. This is true in both tumour-bearing mice (Supplementary Figs. 1k, 1l, 1m) and in C57Bl/6J wildtype mice (Supplementary Figs. 1n, 1o, 1p). Cytotoxicity (Cleaved caspase 3 positivity) was observed in pancreatic tumours (Fig. 1e) but not in normal pancreatic tissues (Supplementary Fig. 1q). Collectively, these data indicate that the dose used in our in vivo studies is reflective of the low dose that kills tumour cells more potently in vitro.

3. While the authors showed that the CR-31-sensitive mRNAs contain relatively structured 5'UTR, it's not clear why the translation of these genes are selectively suppressed in KP organoid compared to N organoid. Is the 5'UTR different or is the loading of eIF4A to 5'UTR different between the organoids?

Response: As noted by the reviewer, we observed radically different effects of CR-31 on the translomes of KP vs N organoids. Our analysis revealed that upon CR-31 treatment, the subset of mRNAs that are translationally activated in KP organoids relative to N organoids revert towards their translational state in N organoids, such that the translome of CR-31-treated KP organoids resembles that of vehicle-treated N organoids (Fig. 2d, Supplementary Table 3). In contrast, translation of this mRNA subset is unaffected upon CR-31 treatment in N organoids (Fig. 2e). This suggests that CR-31 is suppressing hyper-active translation in tumour cells to a basal state but further suppression may be needed to affect the translation in normal cells (indeed Fig. 1b would be consistent with this notion). Heightened sensitivity of PDA cells to CR-31 when compared to normal counterparts may be attributed to differences in the transcriptome between PDA cells and that of normal ductal cells, which in turn leads to alterations in the ability of specific mRNAs to compete for access to the limiting amounts of eIF4F. Alternatively, differential promoter usages in PDA cells could also contribute to this effect. mRNA variants transcribed from alternate promoters and having 5' UTRs⁸ could alter sensitivity and response to eIF4F. Given that human genes have on average more than four transcriptional start sites⁹, the impact of differential promoter usage on translational output is part of our current work and is beyond the scope of this manuscript.

4. The authored showed that CR-31 inhibits the translation of several genes involved in OXPHOS and glucose metabolism. Are the basal transcription or protein levels of these genes different between KP and N organoid?

Response: We have now included an analysis of the basal transcription level of all genes involved in OXPHOS in the revised Fig. 4b. The basal protein level of one of these genes, SDHB, is compared between KP and N organoids by western blot and is presented in revised Supplementary Fig. 6a. As shown in these figures,

although the transcription of these genes remains unchanged, the translation of these mRNAs is upregulated, which results in increased SDHB protein levels in KP compared to N organoids.

The basal protein levels of Hexokinase, Pdh, Pkm, Ldha, GAPDH, and Enolase 1 did not exhibit an increase in protein expression in KP organoids compared to N organoids, as shown in revised **Fig. 4j** and **Supplementary Fig. 7b**. Enolase 2, Enolase 3, SLC2A1, 4, and 6 are upregulated at the basal level in KP compared to N organoids (**Fig. 4j** and **Supplementary Fig. 7b**).

Is the basal OXPHOS and glycolysis different between KP and N organoid? Does CR-31 also suppress the OXPHOS and glycolysis in N organoid?

Response: We have now included data to show that basal levels of OXPHOS and glycolysis are both elevated in KP compared to N organoids. Elevated OXPHOS is reflected through the increase in oxygen consumption (**Supplementary Fig. 6b**) and TCA activity demonstrated by elevated ^{13}C -glucose contribution to citrate in KP organoids (**Fig. 4c**). Increased basal glycolysis in KP organoids is evident by the increase in extracellular acidification rate (**Supplementary Fig. 6k**) and increased ^{13}C -glucose contribution to lactate (**Fig. 4f**). To directly compare the metabolic effects of CR-31 on N and KP organoids, we have now included seahorse analyses of oxygen consumption rates (**Figs. 4d, 4e**) and extracellular acidification rates (**Figs. 4g, 4h**) of both N and KP organoids upon CR-31 treatment. Our data demonstrate heightened sensitivity of KP organoids to CR-31.

5. It will be better to further clarify the effect of CR-31 on glycolysis using ^{13}C -glucose tracing experiment.

Response: ^{13}C -glucose tracing experiments to evaluate the effect of CR-31 on patient-derived pancreatic cancer cells and organoids are presented in revised **Fig. 4f** and **Supplementary Figs. 6l, 6p**. As shown in these figures, ^{13}C -glucose contribution to lactate is reduced in KP organoids upon CR-31 treatment but these changes were not observed in N organoids under the same treatment. The decrease in glycolysis upon CR-31 treatment is evident in all detected glycolytic intermediates (**Supplementary Fig. 6p**), in line with our observation that CR-31 suppresses glycolysis at the level of glucose uptake (**Figs. 4l, 4m, 4n**). We now provide additional data to demonstrate the functional role of the glucose transporter SLC2A6 in PDA cells. Suppression of glycolysis in PDA cells upon CR-31 treatment is in part through translational suppression of the functional glucose transporter SLC2A6 (**Figs. 4o, p, q**).

In addition, what does the ECAR data look like if the authors do baseline normalization in Fig.4e.

Response: Upon baseline normalization, the difference in glycolysis upon CR-31 treatment is no longer evident (**Appendix 3a**). However, the same is also true for the glycolytic difference between KP and N organoids (**Appendix 3b**). Nonetheless, ^{13}C -glucose tracing confirms that glycolysis (^{13}C labelled lactate) is increased in KP compared to N organoids, but decreased to the level of N organoids upon CR-31 treatment (**Fig. 4f**).

Of note, the dose of CR-31 used in Fig.4d is 100nM, in contrast to the 10nM used in other experiments. Please confirm if it's correct or a typo.

Response: We apologize for the confusion. To clarify the differential and dose-dependent response of N and KP organoids to CR-31 in both glycolysis and OXPHOS, we have now revised our figures to present data examining the dose-dependent response of N and KP organoids to CR-31 (**Figs. 4d, 4e, 4g, 4h**).

Appendix 3

Baseline-normalized glycolysis reflected by extracellular acidification rate (ECAR) was measured in KP organoids treated with vehicle or the indicate concentrations of CR-31 for 5 h and following addition of glucose (10 mM), Oligomycin (1 μM) and the glucose analog, 2-deoxyglucose, 2DG (50 mM).

6. The authors tried to claim that the downregulation of Slc2a6 is likely accounted for the decrease in glucose uptake upon CR-31 treatment. As the authors pointed out in the discussion, Slc2a1 and Slc2a3 are the major glucose transporters expressed in tumour cells. A lot more evidence is needed if the authors want to claim that the suppression of Slc2a6 is indeed the mechanism for the decrease of glucose uptake upon CR-31 treatment (please see my general comment in the end). Is Slc2a6 protein level in general higher in tumour cells compare to normal cells? Does Slc2a6 depletion suppress glucose uptake and whether reconstitution of Slc2a6 expression in CR-31 treated cells restores glucose uptake?

Response: The reviewer is correct that the data in our previous manuscript did not provide sufficient evidence that downregulation of SLC2A6 accounts for the decrease in glucose uptake upon CR-31 treatment. Our polysome sequencing data indicate a decrease in the translation of GAPDH, Enolase 2/3 and SLC2A6 upon CR-31 treatment, of these, only SLC2A6 consistently exhibits a decrease in protein expression level (Fig 4j, 4k, Supplementary Fig 7g). No such decrease was observed for the canonical glucose transporters SLC2A1/3/4 (Supplementary Figs. 7b, 7i, 7j). Upon evaluation of protein stability, we observed that SLC2A6 exhibits a shorter half-life than Enolase2/3 and GAPDH (Supplementary Fig. 7h). This likely explains the stronger impact of CR-31 on SLC2A6 than on Enolase2, Enolase 3, and GAPDH even though the translation of all four mRNAs are reduced to similar extents by CR-31 (Fig. 4i). Indeed, after 24 h of CR-31 treatment, we begin to observe moderate decreases in GAPDH, ENOLASE2 and ENOLASE3 expression levels in both murine and patient-derived PDA cells (Fig. 4k, Supplementary Fig. 7g). CRISPR-mediated perturbation of SLC2A6 reduced glucose uptake (Fig. 4o) and glycolysis (Figs. 4p, 4q) in PDA cells, demonstrating that this transporter is functional in PDA cells and that translational suppression of SLC2A6 by CR-31 in part contributes to decreased glucose uptake and glycolysis. Indeed, previous studies have reported elevated expression of SLC2A6 in various cancers, including pancreatic ductal adenocarcinoma^{10,11}.

7. The authors showed that GLS1 inhibition sensitizes PDA cells to CR-31. What's the impact of GLSi alone or in combination with CR-31 on NADPH/NADP ratio and cellular ROS level?

Response: As suggested by the reviewer, we evaluated the impact of CB839 alone and in combination with CR-31 on NADP⁺/NADPH ratio and cellular ROS level. As shown in revised Supplementary Figs. 8b and c, while CR-31 increases the ratio of NADP⁺/NADPH and ROS, combination of CR-31 with CB839 did not further increase NADP⁺/NADPH or ROS.

Does other agents affect NADPH metabolism or ROS regulation, such as BSO, also sensitize cells to CR-31?

Response: Unlike glutaminase inhibitors, (BPTES, Fig. 5g; CB839, Figs. 5h, 5i, 5j), the glutathione synthesis inhibitor BSO did not sensitize cells to CR-31 (Fig. 3l).

Will the cytotoxicity of CR-31 or the CR-31/CB839 combination be rescued with ROS scavenger?

Response: The cytotoxicity of CR-31 alone (Fig. 3m) or in combination with CB839 (Supplementary Fig. 8d) cannot be rescued by the ROS scavenger N-acetylcysteine. In conjunction with the observation that combined treatment of CR-31 with CB839 does not further increase ROS (Supplementary Figs. 8b, 8c), and that other agents that affect ROS regulation eg BSO, did not sensitize cells to CR-31 (Fig. 3l), we conclude that the increased cytotoxicity observed upon combined inhibition of eIF4A and glutaminase is not redox-dependent. We noted that the fatty acid synthesis inhibitor Orlistat was able to modestly sensitize PDA cells to CR-31 treatment (Supplementary Fig. 8e). As glutamine reductive carboxylation is the major pathway of citrate formation and subsequent fatty acid synthesis, these results suggest that sensitization of PDA cells to CR-31 through co-suppression of glutaminase activity occurs through redox-independent mechanisms possibly through reducing carbon flow towards fatty acid synthesis.

8. In general, the authors provided a comprehensive characterization of the impact of CR-31 treatment on cellular metabolism. In addition, the authors also observed changes in the translation of multiple genes in those metabolism pathways, such as ETC, glucose uptake and glutamine metabolism. It's important for the authors to present these data to readers. However, mechanistic speculations should be avoided. For example, is the downregulation of Slc2a6 by CR-31 potent enough to suppress glucose uptake in PDA cells?

Response: As discussed above, sgSLC2A6-expressing PDA cells exhibit a decrease in glucose uptake and also glycolysis when compared to sgROSA-expressing counterparts (Figs. 4o, p, q) Our data support that SLC2A6 is a functional glucose transporter in PDA cells and that translational suppression of this transporter in part contributes to decreased glucose metabolism in these cells.

Does the induction of reductive carboxylation have anything to do with the change in NADPH/NADP ratio and ROS level upon CR-31 treatment? In my opinion, the observations are interesting enough by themselves although the data are mostly correlative at current state. There's no need for the authors to offer mechanistic explanations to every observation in one manuscript. But the authors should consider re-organizing the data description and clarifying the discussion to avoid potentially misleading the readers.

Response: We thank the reviewer for this comment. We agree that our data do not exclude the possibility that other mechanisms may also exist to perturb NADP⁺/NADPH ratios to elevate ROS in CR-31-treated cells. Indeed, following the recommendation of reviewer #1, we reanalyzed our polysome sequencing data using GAGE analysis and found that in addition to oxidative phosphorylation, we observed pathway enrichment for mRNAs involved in glutathione metabolism in KP relative to N organoids (Supplementary Fig. 5, Fig. 3a). We also observed that the translation of these mRNAs is selectively downregulated in KP organoids upon CR-31 treatment (Supplementary Fig. 5, Fig. 3a). These results suggest that maintenance of redox homeostasis is an oncogenic activity of the eIF4F complex that can be targeted using CR-31. This observation is consistent with a previous study showing that eIF4E is critical for select translation of mRNAs that control intracellular ROS levels during cellular transformation¹². Examination of the impact of CR-31 in KP organoids is now presented in revised Fig. 3.

Appendix 4

Ratio of NAD⁺/NADH in primary human pancreatic cancer cells with vehicle or 100 nM CR-31 for 16 h. Data are mean ± S.D. (n=3, Student's t test). NS= not significant. * = p<0.05 compared to DMSO treatment.

With regards to other mechanisms that may induce reductive carboxylation, it has been shown previously that reductive carboxylation is an important feature of cells proliferating in hypoxia¹³. Decreased NAD⁺/NADH ratio in the mitochondria under hypoxic conditions is expected to increase levels of NADPH through the activity of the mitochondrial transhydrogenase¹⁴, which in turn could support reductive carboxylation. However, we did not observe any decrease in NAD⁺/NADH ratios in PDA lines tested. In fact, we observed an increase in NAD⁺/NADH ratios in CR-31-treated cells (Appendix 4), likely due to CR-31 mediated increase in ROS. We also did not observe any transcriptional increase in common HIF1α target genes such as SLC2A1 and LDHA (Supplementary Fig. 7a). We reasoned that hypoxia or elevated HIF1 activity are unlikely to account for elevated reductive carboxylation observed in CR-31 treated cells.

Reviewer #3 (Remarks to the Author):

Chan, Robert, Oertlin, Kapeller-Libermann, and colleagues study the role of eIF4A-mediated translation in regulating the metabolic pathways of human and murine pancreatic cancer cells. The study uses multiple in

vitro systems with some in vivo validation to argue that inhibition of eIF4A-controlled translation by CR-31 specifically alters the metabolic state of tumour cells resulting in increased oxidative stress and reliance on glutamine for compensation in the synthetic and redox pathways.

The idea is conceptually interesting and the experiments provide some proof-of-principle support. I would ask for the following issues to be addressed:

1. The authors use normal ducts and duct-derived organoids as their comparison control group. This comparison is used to argue for CR-31's efficacy and to identify the changes in polysome-associated RNAs. Given that the KP organoids are actually acinar cell-derived in these mice, this control makes less sense to me. Why were these studies performed that way and do the authors have any data from native acinar cells? If not, could the authors at least replicate the viability and protein synthesis control experiments with acinar cell controls.

Response: While pancreatic acinar cells can give rise to PDA through the process of acinar-ductal metaplasia, pancreatic ductal cells do represent an effective cell of origin for PDA^{15,16}. In fact, it has been shown that ductal cells are primed to form carcinoma in situ that become invasive PDA in the presence of Kras and Trp53 mutations, while acinar cells with the same mutations require a prolonged period of transition or reprogramming to initiate PDA¹⁶. Since identical oncogenic drivers could trigger PDA originating from both ductal and acinar cells with similar histology¹⁷, we selected normal proliferating ducts as an experimental system to probe for differences in cellular response to eIF4A inhibition in KRAS mutant PDA cells relative to KRAS wild-type, proliferating ductal counterparts cultured under the same conditions.

*Nonetheless, the reviewer raises a very good point that our approach indeed does not provide a comprehensive comparison of PDA cells relative to the normal exocrine pancreas, which includes both ductal and acinar cells. We attempted to culture acinar cells to examine the effect of CR-31 but unfortunately were not successful. Cultivating acinar cells in vitro has been challenging due to the intrinsic sensitivity of pancreatic tissue to experimental manipulation due to the high content in glycolytic, proteolytic, and lipolytic enzymes, which will digest the pancreatic tissue when they are released during the isolation of pancreatic cells. Given the above, we sought to address the impact of CR-31 treatment on PDA relative to normal cells in vivo. When treated with CR-31 for 7 days, we noted an increase in cleaved caspase 3 (CC3) staining in PDA cells when compared to vehicle-treated controls (**Fig. 1e**). However, when C57Bl/6J mice were given the same treatment for up to 12 days, no induction of CC3 was observed in the normal pancreas (**Supplementary Fig. 1q**). Likewise, no change in body weight, circulating liver enzymes, or gross histological changes in normal tissues were noted in mice treated with CR-31 compared to vehicle controls (**Supplementary Figs. 1k-q**). Collectively, our data indicate that neoplastic cells exhibit a higher sensitivity to eIF4A inhibition than normal, non-transformed cells both in vitro and in vivo.*

2. Was there any toxicity seen in the mice during the tumour treatments? Were mouse weights recorded during these experiments and could this data be provided?

*Response: In a previous study, mice receiving repeated daily injections at 0.2 mg kg⁻¹ CR-31 (same dose used in current study) exhibited no observable systemic toxicity and CR-31 did not display toxicity toward cells of hematopoietic origin¹. This is consistent with our new data showing that body weight, liver enzymes, pancreatic enzymes, and gross histology are not changed in CR-31-treated mice compared to vehicle control. This was true in both tumour bearing mice (**Supplementary Figs. 1k, 1l, 1m**) and C57Bl/6J wild-type mice (**Supplementary Figs. 1n, 1o, 1p**). Cytotoxicity (Cleaved caspase 3 positivity) was observed only in pancreatic tumours but not in the normal pancreas (**Supplementary Fig. 1q**).*

3. The statements revolving around CR-31's selectivity specifically for pancreatic cancer cells are too strong

in this manuscript. The authors do not include nearly enough controls to really prove that. I would tone down the language. Example of this can be found on page of the Results – start of the mRNA modulation section.

Response: We thank the reviewer for bringing this to our attention. We have now revised the main body of our manuscript to more clearly present the message that pancreatic cancer cells are more sensitive to eIF4A inhibition using CR-31 but the effect is not exclusive to cancer cells.

4. Could the authors clarify in their Methods or Results somewhere what they mean by “four pairs of murine N and KP organoids”? Was this a true paired design, where “normal” pancreas was obtained from KP mice or is this simply four normal mice and four KP mice?

Response: N and KP organoids were derived from ductal isolates of C56Bl/6J wild-type mice and KrasG12D;p53R172H;PdxCre mice, respectively. We have now clarified this in our methods sections.

5. The section on 5'UTR region composition and structure is interesting although a bit ancillary to the rest of the manuscript. I would caution the authors about the statement made regarding similarity to the general structure features found in the Rubio manuscript. In that paper, the silvestrol-regulated transcripts with lower translation had longer 5'UTRs and lower GC content. This is opposite to what is seen in the CR-31 sensitive transcripts, which have shorter 5'UTR with a higher GC content.

Response: We thank the reviewer for pointing this out. Indeed, our study supports the notion that rocaglates may suppress translation of distinct subsets of transcripts in a context and/or agent specific manner. We now make these differences clearer by pointing out that in contrast to RocA, we do not observe an increased proportion of polypurine motifs in transcripts translationally suppressed by CR-31 in KP organoids. (Supplementary Fig. 4a). We have also removed our comments regarding the similarity of our dataset to the general structures found in the Rubio manuscript.

6. I generally like the experimental work in the carbon metabolism section. I think the data regarding the ETC inhibition and oxygen consumption are solid. I would again caution the authors regarding their language for the results of the shRNA eIF4A experiment. This experiment phenocopies the CR-31 data (as the authors rightly point out in their Discussion). It is not a proof of CR-31's mechanism specificity through eIF4a inhibition.

Response: We thank the reviewer for bringing this to our attention. We have now revised the main body of our manuscript to clarify our message. Biochemical assays using recombinant eIF4A^{1,18,19}, affinity chromatography experiments using immobilized epi-silvestrol²⁰, chemogenomic profiling in yeast²¹, and the characterization of rocaglate-resistant eIF4A alleles²² have identified eIF4A as a predominant target of rocaglates. We now include data from a cell-free assay to show that CR-31 decreases the dissociation rate of eIF4A from RNAs in the presence or absence of ATP (Supplementary Fig. 1d), thus confirming its inhibitory effect on eIF4A^{1,23,2}. Results from our shRNA eIF4A experiment support these previous observations to show that eIF4A availability contributes to the metabolic phenotypes observed herein.

7. The glycolysis inhibition data is less consistent. The extracellular acidification rate is certainly reduced. However, the protein level data does not suggest any significant changes of enzyme levels. The steady state labeling with ¹³C-glucose results are also hard to interpret. The data shows a decrease in labeling of glycolytic intermediates, but this is just 5% change in most of the intermediates measured here. It is difficult to square that with the acidification and lactate level data unless we know how long the ¹³C-glucose was in the system given that glycolytic flux is a fast process in pancreatic tumour cells. Could the authors provide information on how long they exposed the cells to ¹³C-glucose and more specifically discuss other carbon sources in their system in their Methods or Results?

Response: In the revised manuscript, we provide data to show that CR-31 treatment decreases glucose uptake in vitro (Fig. 4l) and in vivo (Fig. 4m, 4n). The reviewer is correct that no protein expression decrease was observed in various enzymes involved in glycolysis (Enolase1/2/3, GAPDH, LDHA, PKM, Pdh, Hk1/2 etc). However, consistent with our polysome sequencing data, we noted a decrease in the expression level of the glucose transporter SLC2A6 upon CR-31 treatment. No such decrease was observed for the canonical glucose transporters SLC2A1/3/4 (Supplementary Figs. 7b, 7i, 7j). Upon evaluation of protein stability, we observed that SLC2A6 exhibits a shorter half-life than Enolase and GAPDH (Supplementary Fig. 7h). This likely explains the stronger impact of CR-31 on SLC2A6 than on Enolase2, Enolase 3, and GAPDH even though the translation of all four mRNAs are reduced to similar extents by CR-31 (Fig. 4i). Indeed, after 24 h of CR-31 treatment, we begin to observe moderate decreases in GAPDH, ENOLASE2 and ENOLASE3 expression in both murine and patient-derived PDA cells (Fig. 4k, Supplementary Fig. 7g). CRISPR-mediated perturbation of SLC2A6 reduced glucose uptake (Fig. 4o) and glycolysis (Figs. 4p, 4q) in PDA cells, demonstrating that this transporter is functional in PDA cells.

For ¹³C-glucose tracing experiments, ¹³C media was used to replace ¹²C media 2 hours before metabolite extraction. Given the high rate of glycolysis, the differences we observed in most glycolytic intermediates are small. A time course ¹³C-tracing experiment would have been ideal to capture the kinetic differences in the glycolytic flux of cells upon CR-31 treatment. As recommended by the reviewer, this information is now included in the methods and also the results sections of our manuscript to improve clarity.

8. As a sidenote, the way of displaying the data of the stable metabolic labeling is at best unintentionally misleading with the choice of Y axis ranges to maximize and magnify the apparent difference between the treatments. If the axes were ranged from 0, the small magnitude of the changes would be more faithfully demonstrated.

Response: We apologize for the confusion. We have revised our figure legends as recommended by the reviewer.

My overall take from the paper is that regulation of translation certainly affects tumour cell metabolism. The data to support this as a possible therapeutic approach are much less clear (partly due to the choice of the control using “normal ducts”) and the authors should temper their comments here.

Response: In the revised manuscript, we have incorporated the reviewer’s suggestion in the results and discussion sections.

I found the experimental results interesting by themselves without having to resort to the “sales pitch” of treatment. The glycolysis experiments need to be clarified. Thank you for allowing me to review this work and I look forward to the revision!

Response: We thank the reviewer for their positive and very insightful suggestions for our manuscript.

References

- 1 Rodrigo, C. M., Cencic, R., Roche, S. P., Pelletier, J. & Porco, J. A. Synthesis of Rocaglamide Hydroxamates and Related Compounds as Eukaryotic Translation Inhibitors: Synthetic and Biological Studies. *Journal of medicinal chemistry* **55**, 558-562, doi:10.1021/jm201263k (2012).
- 2 Iwasaki, S., Floor, S. N. & Ingolia, N. T. Rocaglates convert DEAD-box protein eIF4A into a sequence-selective translational repressor. *Nature* **534**, 558+, doi:10.1038/nature17978 (2016).
- 3 Rubio, C. A. *et al.* Transcriptome-wide characterization of the eIF4A signature highlights plasticity in translation regulation. *Genome Biol* **15**, 476, doi:10.1186/s13059-014-0476-1 (2014).
- 4 Wolfe, A. L. *et al.* RNA G-quadruplexes cause eIF4A-dependent oncogene translation in cancer. *Nature* **513**, 65-70, doi:10.1038/nature13485 (2014).
- 5 Delmore, J. E. *et al.* BET Bromodomain Inhibition as a Therapeutic Strategy to Target c-Myc. *Cell* **146**, 903-916, doi:10.1016/j.cell.2011.08.017 (2011).
- 6 Larsson, O. *et al.* Eukaryotic translation initiation factor 4E induced progression of primary human mammary epithelial cells along the cancer pathway is associated with targeted translational deregulation of oncogenic drivers and inhibitors. *Cancer Res* **67**, 6814-6824, doi:10.1158/0008-5472.CAN-07-0752 (2007).
- 7 Larsson, O. *et al.* Distinct perturbation of the translome by the antidiabetic drug metformin. *Proc Natl Acad Sci U S A* **109**, 8977-8982, doi:10.1073/pnas.1201689109 (2012).
- 8 Livingstone, M. *et al.* Assessment of mTOR-Dependent Translational Regulation of Interferon Stimulated Genes. *PloS one* **10**, e0133482, doi:10.1371/journal.pone.0133482 (2015).
- 9 Consortium, F. *et al.* A promoter-level mammalian expression atlas. *Nature* **507**, 462-470, doi:10.1038/nature13182 (2014).
- 10 Godoy, A. *et al.* Differential subcellular distribution of glucose transporters GLUT1-6 and GLUT9 in human cancer: ultrastructural localization of GLUT1 and GLUT5 in breast tumour tissues. *Journal of cellular physiology* **207**, 614-627, doi:10.1002/jcp.20606 (2006).
- 11 Byrne, F. L. *et al.* Metabolic vulnerabilities in endometrial cancer. *Cancer Res* **74**, 5832-5845, doi:10.1158/0008-5472.CAN-14-0254 (2014).
- 12 Truitt, M. L. *et al.* Differential Requirements for eIF4E Dose in Normal Development and Cancer. *Cell* **162**, 59-71, doi:10.1016/j.cell.2015.05.049 (2015).
- 13 Wise, D. R. *et al.* Hypoxia promotes isocitrate dehydrogenase-dependent carboxylation of alpha-ketoglutarate to citrate to support cell growth and viability. *Proc Natl Acad Sci U S A* **108**, 19611-19616, doi:10.1073/pnas.1117773108 (2011).
- 14 Rydstrom, J. Mitochondrial NADPH, transhydrogenase and disease. *Biochimica et biophysica acta* **1757**, 721-726, doi:10.1016/j.bbabi.2006.03.010 (2006).
- 15 Bailey, J. M. *et al.* p53 mutations cooperate with oncogenic Kras to promote adenocarcinoma from pancreatic ductal cells. *Oncogene* **35**, 4282-4288, doi:10.1038/onc.2015.441 (2016).
- 16 Lee, A. Y. L. *et al.* Cell of origin affects tumour development and phenotype in pancreatic ductal adenocarcinoma. *Gut*, doi:10.1136/gutjnl-2017-314426 (2018).
- 17 Ferreira, R. M. M. *et al.* Duct- and Acinar-Derived Pancreatic Ductal Adenocarcinomas Show Distinct Tumour Progression and Marker Expression. *Cell reports* **21**, 966-978, doi:10.1016/j.celrep.2017.09.093 (2017).
- 18 Cencic, R. *et al.* Antitumour activity and mechanism of action of the cyclopenta[b]benzofuran, silvestrol. *PloS one* **4**, e5223, doi:10.1371/journal.pone.0005223 (2009).
- 19 Bordeleau, M. E. *et al.* Therapeutic suppression of translation initiation modulates chemosensitivity in a mouse lymphoma model. *Journal of Clinical Investigation* **118**, 2651-2660, doi:10.1172/Jci34753 (2008).
- 20 Chambers, J. M. *et al.* Synthesis of Biotinylated Episilvestrol: Highly Selective Targeting of the Translation Factors eIF4A/II. *Organic letters* **15**, 1406-1409, doi:10.1021/ol400401d (2013).
- 21 Sadlish, H. *et al.* Evidence for a functionally relevant rocaglamide binding site on the eIF4A-RNA complex. *ACS Chem Biol* **8**, 1519-1527, doi:10.1021/cb400158t (2013).

- 22 Chu, J. *et al.* CRISPR-Mediated Drug-Target Validation Reveals Selective Pharmacological Inhibition of the RNA Helicase, eIF4A. *Cell reports* **15**, 2340-2347, doi:10.1016/j.celrep.2016.05.005 (2016).
- 23 Chu, J. & Pelletier, J. Targeting the eIF4A RNA helicase as an anti-neoplastic approach. *Bba-Gene Regul Mech* **1849**, 781-791, doi:10.1016/j.bbagr.2014.09.006 (2015).

REVIEWERS' COMMENTS:

Reviewer #1 (Remarks to the Author):

The authors have responded well to previous comments. That said, in part owing to the large amount of data presented, there are still a few concerns, but none that are likely to require additional experimentation.

Concerns

1. Introduction – eIF4A appears to be required for both the binding of mRNAs to the 43S PIC as well as scanning and both steps are ATP-dependent. (line70, 71)
2. line 103 – “in the presence and absence of ATP” – This is incorrect as eIF4A does not bind RNA in the absence of ATP. The authors used ADPNP.
3. Inhibited or stimulated – it would be helpful to indicate the degree of inhibition or stimulation (i.e. 27% inhibition).
4. Line 132 – In the mid-1980’s, Dr. Robert Thach’s group generated a series of papers that described the competition between mRNAs. In essence, the least competitive mRNAs would generally be silenced with a reduction in eIF4F activity. In a future consideration, one might ask whether the mRNAs that are downregulated by CR-31 are less efficiently translated under normal conditions (i.e. fewer ribosomes per 1000 nucleotides).
5. Line 177-184 – The data in Supplemental Figure 4 show a slight difference in both the 5’ UTR length and the GC content of the downregulated mRNAs. However, it is not clear that the difference is statistically significant. Thus, the statement that “ CR-31 ...effectively reprograms the translome of KP organoids...” appears to be an overstatement.
6. Lines 220-230 – The monitoring of OCR is at 5 hours, but the measurement of phosphor-AMPK is at 24-36 hours. Are these really occurring in the same time frame?
7. The impression given is that the eIF4A reduction caused suppression of oxidative phosphorylation is the same in human and mice. However, while the reduction is 2-fold in mice, the reduction is only 10-20% in humans. This actually puts at some risk any mouse studies if this is to be truly representative of humans as well.
8. Line “...global levels of intracellular and secreted lactate were dramatically reduced.” This is true for KP (32 20), but for the patient samples the reduction is much less (48 40). Thus, the patient samples are not as dramatic.
9. The affects of CB839 and Orlistat – both of these compounds appear to have a similar affect on viability with a reduction vs CR-31 only in a very narrow concentration range (around 6.25 nM). Is this likely to be useful therapeutically?
10. By this reviewer’s calculation, 2 mg per kg treatment with CR-31 would be a concentration that is about 400 nM (assuming one kg is equivalent to a liter). Yet the effective and tested levels of CR-31 used are more around 10 nM. Does this reflect drug metabolism occurring or is there another explanation?

Reviewer #2 (Remarks to the Author):

The authors have addressed my concerns and I have no further comment.

Reviewer #3 (Remarks to the Author):

Chan and colleagues now present the first revision of their manuscript in which they describe the activity of the rocaglate CR-31 on the eIF4A oncogenic translational programs in Kras-driven pancreatic cancer using both murine and human tumor tissue. The manuscript has been reorganized to highlight new data. The authors demonstrate a Kras-associated increase in

translation with alterations in the metabolic and redox systems. They convincingly show the effects of CR-31 on oxidative phosphorylation, glycolysis, and the glutamine metabolism. The combination of CR-31 as a translation inhibitor with other metabolic inhibitors is presented as a potential therapeutic avenue for pancreatic cancer in the future.

The manuscript's reorganization and additional data have significantly clarified it and I enjoyed reading the revision. The authors have addressed my concerns to my satisfaction and I support its publication in Nature Communications. Thanks for allowing me to participate in its review and congratulations to the authors on a very nice and complete piece of work.

We thank the reviewer for their constructive comments and suggestions. We have revised our manuscript to address the following concerns:

1. Introduction – eIF4A appears to be required for both the binding of mRNAs to the 43S PIC as well as scanning and both steps are ATP-dependent. (line70, 71)

Response: We have revised the manuscript to incorporate these points.

2. line 103 – “in the presence and absence of ATP” – This is incorrect as eIF4A does not bind RNA in the absence of ATP. The authors used ADPNP.

Response: We have made this correction in the revised manuscript.

3. Inhibited or stimulated – it would be helpful to indicate the degree of inhibition or stimulation (i.e. 27% inhibition).

Response: We have revised the manuscript to incorporate these points.

4. Line 132 – In the mid-1980’s, Dr. Robert Thach’s group generated a series of papers that described the competition between mRNAs. In essence, the least competitive mRNAs would generally be silenced with a reduction in eIF4F activity. In a future consideration, one might ask whether the mRNAs that are downregulated by CR-31 are less efficiently translated under normal conditions (i.e. fewer ribosomes per 1000 nucleotides).

Response: We thank the reviewer for this fantastic suggestion, we will test this in our follow up study.

5. Line 177-184 – The data in Supplemental Figure 4 show a slight difference in both the 5’ UTR length and the GC content of the downregulated mRNAs. However, it is not clear that the difference is statistically significant. Thus, the statement that “ CR-31 ...effectively reprograms the translome of KP organoids...” appears to be an overstatement.

Response: We regret that we were unclear in the description of the results. The statement refers to changes in translational efficiencies, not changes in 5’ UTR features alone. We have clarified this in line 273.

6. Lines 220-230 – The monitoring of OCR is at 5 hours, but the measurement of phosphor-AMPK is at 24-36 hours. Are these really occurring in the same time frame?

Response: We monitored OCR at 5 hours to assess the immediate effects of CR-31 on the translation of ETC components. These effects persist up to 24 and 36 hours after treatment. Suppressed ETC activity results in an increase in AMP/ATP ratio, resulting in a conformational shift that makes Thr172 on AMPK to be more amenable to phosphorylation by LKB1. Activated AMPK subsequently restores energy homeostasis by promoting catabolic pathways of ATP production. Thus, we monitor the activation of AMPK at later timepoints as a means to gauge the level of energy stress induced as a consequence of OxPHOS inhibition by CR-31. Similar timepoints are used in other studies to evaluate the impact of other OxPHOS inhibitors such as metformin on the induction of energy stress.

7. The impression given is that the eIF4A reduction caused suppression of oxidative phosphorylation is the same in human and mice. However, while the reduction is 2-fold in mice, the reduction is only 10-20% in humans. This actually puts at some risk any mouse studies if this is to be truly representative of humans as well.

Response: We agree with the reviewer that mouse studies do not fully represent the biology of human cells. Thus, in our manuscript, we present data in both mouse and human settings whenever possible. The degree of acute inhibition of oxPHOS using CR-31 occurs to a similar degree in both mouse and human cells, as shown in Figure 4d and Supplementary Figure 7e.

8. Line “...global levels of intracellular and secreted lactate were dramatically reduced.” This is true for KP

(32 20), but for the patient samples the reduction is much less (48 40). Thus, the patient samples are not as dramatic.

Response: We have revised our manuscript to the following: “When KP organoids (Fig. 4f, Supplementary Figs. 8e, 8f) or patient-derived PDA cells (Fig. 4f) were treated with CR-31, global levels of intracellular and secreted lactate were reduced.”

9. The affects of CB839 and Orlistat – both of these compounds appear to have a similar affect on viability with a reduction vs CR-31 only in a very narrow concentration range (around 6.25 nM). Is this likely to be useful therapeutically?

Response: We agree with the reviewer that the effects of CB839 and Orlistat to reduce viability when combined with CR-31 occurred at a narrow concentration range. We believe that these data provide a prototype to demonstrate the importance of eIF4A in translational control of pancreatic tumour metabolism and as a therapeutic target against PDA. Our future studies will involve investigating additional inhibitors targeting the pathway of reductive carboxylation and fatty acid synthesis. We hope these future endeavours will identify a combination strategy that can be clinically useful.

10. By this reviewer’s calculation, 2 mg per kg treatment with CR-31 would be a concentration that is about 400 nM (assuming one kg is equivalent to a liter). Yet the effective and tested levels of CR-31 used are more around 10 nM. Does this reflect drug metabolism occurring or is there another explanation?

Response: To achieve full drug efficacy, it is not uncommon to target effective plasma concentrations that are well higher than the EC50 (e.g. the cellular concentration required to see half effect). Further, in vitro effective concentrations rarely directly extrapolate to in vivo doses because plasma concentrations post-administration are usually much lower than the administered dose divided by blood volume. For intraperitoneal administration this is caused by a number of factors, which include metabolism and other mechanisms of drug clearance, as well as the compound's stability, degree of distribution to tissues, binding affinity for plasma proteins, etc.